# Toward Evaluative Thinking: Meta Policy Optimization with Evolving Reward Models

## Abstract

Reward-based alignment methods for large language models (LLMs) face two key limitations: vulnerability to reward hacking, where models exploit flaws in the reward signal; and reliance on brittle, labor-intensive prompt engineering when LLMs are used as reward models. We introduce **Meta Policy Optimization (MPO)**, a framework that addresses these challenges by integrating a meta-reward model that dynamically refines the reward model's prompt throughout training. In MPO, the meta-reward model monitors the evolving training context and continuously adjusts the reward model's prompt to maintain high alignment, providing an adaptive reward signal that resists exploitation by the policy. This meta-learning approach promotes a more stable policy optimization, and greatly reduces the need for manual reward prompt design. It yields performance on par with or better than models guided by extensively hand-crafted reward prompts. Furthermore, we show that MPO maintains its effectiveness across diverse tasks, from essay writing to mathematical reasoning, without requiring specialized reward designs. Beyond standard RLAIF, MPO's meta-learning formulation is readily extensible to higher-level alignment frameworks. Overall, this method addresses theoretical and practical challenges in reward-based RL alignment for LLMs, paving the way for more robust and adaptable alignment strategies. The code and data can be accessed at: https://anonymous.4open.science/r/mpo-CD3B

## 1 Introduction

*Good thinkers must use another high-level thinking skill, namely, analysis and awareness of one's own thinking—or metacognition (Buckley et al., 2015; Lord et al., 1979).*

Recent advancements in reinforcement learning (RL) for large language model (LLM) training have marked a shift from tasks that prioritize human-like response generation (Ziegler et al., 2020; Stiennon et al., 2020; Ouyang et al., 2022b) to those that emphasize structured reasoning, such as mathematics and programming (Shao et al., 2024a; DeepSeek-AI et al., 2025). Traditionally, human-aligned answering relies on reward models trained using binary comparison data, whereas structured reasoning tasks focus more on verifying the correctness of final answers or logical processes based on objective ground truth. However, many real-world challenges demand a synthesis of both approaches—requiring models to integrate human-aligned judgment with rigorous reasoning. This introduces significant complexity, as such tasks often lack objectively verifiable "golden answers," yet still necessitate coherent and justifiable reasoning.

A scalable approach to subjective evaluation is to use an LLM as a judge—an approach commonly referred to as reinforcement learning with AI feedback (RLAIF)—where a fixed prompt is used to assess model performance on specific tasks. However, this method introduces three key challenges. **First, calibration**: effective scoring requires evaluation criteria that are sufficiently detailed and carefully calibrated to the quality and variability of the policy model's outputs. Overly granular feedback on poor responses may be too complex to meaningfully guide improvement, while feedback that is too coarse or simplistic may fail to drive meaningful behavioral change in the model. **Second, reward hacking:** it is well-known that when LLMs are trained with imperfect reward functions—as is often the case with LLM-as-a-judge or reward models—they may learn to exploit shortcuts that maximize reward without genuinely improving output quality or alignment with human values (Amodei et al., 2016b; Everitt et al., 2021; Langosco et al., 2022; Pan et al., 2022). These shortcuts can lead to responses that are formally rewarded yet misaligned with human intent or utility. **Third, prompt-engineering overhead:** substantial manual effort is often required for prompt engineering when generating training data for reward models or employing LLMs as proxy reward functions. This process introduces scalability bottlenecks and limits automation in alignment pipelines.

To address these issues, this work introduces **Meta Policy Optimization (MPO)** (Figure 1), a framework that augments existing reward-based RLAIF pipeline by adding **a meta reward model**. Unlike a traditional reward model that simply scores the policy's output based on a fixed prompt, the meta reward model monitors the evolving training landscape and adjusts or refines the prompt used by the standard reward model. Our design of MPO is inspired by the psychological concept of metacognition—the process of becoming aware of and reflecting on one's own thinking (Flavell, 1979)—and its central role in **evaluative thinking**, a reflective, evidence-driven cognitive process that involves questioning, analyzing, and interpreting information to guide decision-making and continuous learning (Buckley et al., 2015).

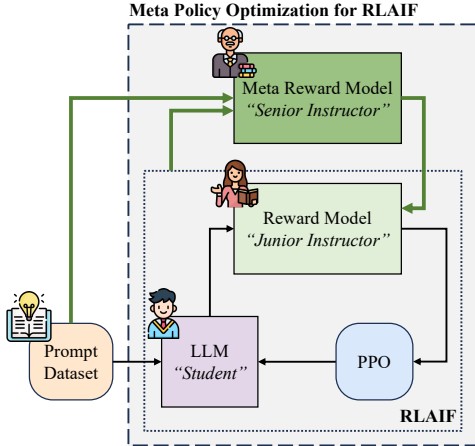

Figure 1: In standard RLAIF, the reward model used during proximal policy optimization (PPO) remains fixed throughout RL alignment. In contrast, MPO framework (in green) introduces a meta reward model that dynamically evolves the reward model based on the current training context, including the task prompt, sampled generations with associated scores, and the latest evaluation prompt. MPO leverages this contextual information to iteratively refine the evaluation prompt, enabling more adaptive and effective alignment.

Metacognitive awareness and control are essential to this process, enabling individuals to monitor reasoning, detect biases, and refine strategies through task assessment and reflection. Cognitive science research further supports this by showing that deliberate reflection fosters deeper, more robust learning (McCormick, 2003; Metcalfe & Kornell, 2005; Veenman et al., 2006; Efklides, 2006). By carrying these principles into RL-based alignment for LLMs, we unlock several advantages:

- **Greater stability in RLAIF training**: MPO dynamically adjusts reward model prompts to deliver context-sensitive scoring criteria based on the policy model's performance, while also mitigating reward hacking—exploitation behaviors often seen in fixed-prompt setups.
- **Reduced prompt engineering burden**: MPO iteratively refines and expands existing prompts within a single training cycle, eliminating repeated manual intervention.
- **Flexible and general framework**: MPO can be used across diverse tasks (see Section 3.3) without major modifications to the training procedure.

## 2 META POLICY OPTIMIZATION

As noted in the introduction, our MPO approach draws inspiration from the fields of metacognition and evaluation. We elaborate on this conceptual motivation, then present a formal reinforcement learning formulation of MPO, followed by a detailed description of its implementation steps.

### 2.1 EVALUATIVE THINKING

Evaluative thinking (ET) (Buckley et al., 2015) is the intentional process of analyzing, interpreting, and assessing information to support thoughtful decision-making, playing a critical role in evaluation capacity building (ECB). However, current reward models in RL for LLM lack such metacognitive control. These models are typically trained on static human preference datasets or guided by fixed rubric prompts, and they remain unchanged during training. As the policy improves, the static reward model tends to collapse nuanced improvements into a coarse label—good enough—allowing early blind spots to persist and go uncorrected.

Motivated by the relationship between Evaluative Thinking (ET) and Evaluation Capacity Building (ECB)—where ET supports ECB by enhancing metacognition—we propose a Meta Reward Model (MRM) that guides the reward model to develop evaluative metacognition and become a more effective scorer. Specifically, the MRM follows the core principles of ET: evidence gathering, questioning, and reflective judgment (see Section 2.3 and Figure 3). Our MPO framework operationalizes ET by enabling the MRM to refine RM's observational partitions over time. In this setup, the reward model improves through on-policy learning driven by the metacognitive signals of the MRM.

**Remark 1** (Depth and Breadth of ET). We posit that ET in the context of reinforcement learning for LLMs can be understood along two orthogonal dimensions: *depth* and *breadth*. This framework echoes Edward de Bono's celebrated distinction between vertical and lateral thinking (De Bono, 1971). Intuitively, depth corresponds to sequential, instance-specific reasoning—reflecting the degree of logical inference and deliberation required to evaluate a single case. In contrast, breadth captures

the ability to generalize across varied instances, recognizing recurring patterns or abstract principles that inform evaluation in novel contexts. As illustrated in Figure 2, tasks such as mathematical reasoning exemplify vertical (deep) thinking, as they involve multi-step, case-specific deductions. In contrast, open-ended tasks like essay writing align more with lateral (broad) thinking, requiring evaluative generalization across diverse prompts. To empirically explore these dimensions, we evaluate four representative tasks—mathematical reasoning, ethical reasoning, summarization, and essay writing—each occupying a distinct region of the ET depth–breadth space in our experiments.

## 2.2 TIME-VARYING OBSERVATIONS AND REWARDS BY EVOLVING REWARD MODEL

While ECB through ET provides a foundation for improving the RM, it does not fully capture the dynamics of RL, where learning is driven by signals from an ECB-enhanced RM. To address this gap, we introduce a mathematical framework that formalizes how the RM evolves under the influence of ET and how this evolving RM can be integrated into the training process of LLMs in Appendix B.

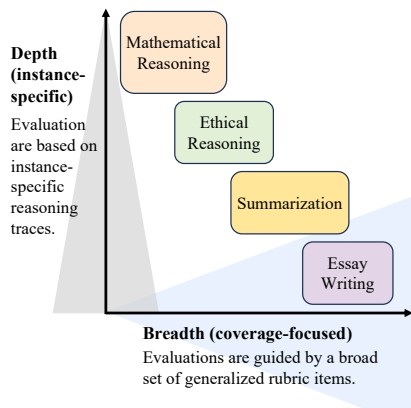

Figure 2: *Depth* and *Breadth* of ET.

### 2.2.1 META POLICY OPTIMIZATION FRAMEWORK

Employing a single static LLM scorer corresponds to a fixed observation partition, potentially too coarse to accurately capture nuanced reward differences. Such coarse partitioning groups many distinct states (e.g., texts or dialog histories) into overly coarse categories, leading to averaged rewards of the form:

$$R(o) = \mathbb{E}_{s \in \mathcal{O}_o}[r(s)],$$

which obscure state-specific details crucial for precise policy optimization (See the definition of $\mathcal{O}_o$ in Appendix B). Consequently, a static scoring mechanism struggles to converge towards the ground-truth reward $r(s)$, limiting its ability to capture subtle, high-dimensional, or evolving reward criteria. In contrast, a meta policy framework addresses this limitation by adaptively refining observation partitions over successive iterations. Formally, the meta-rewarding process introduces progressively finer partitions:

$$\{\mathcal{O}_{o,t}\}_{o \in \Omega_t}, \quad \text{where} \quad \mathcal{O}_{o,t} \subseteq \mathcal{O}_{o',t-1} \quad \text{for some } o', t > 1.$$

This iterative refinement enables increasingly discriminative reward signals:

$$R_t(o) = \mathbb{E}_{s \in \mathcal{O}_{o,t}}[r(s)],$$

that better capture subtle variations in the state space. By adaptively partitioning the observation sets—splitting larger, coarse categories into smaller, targeted subsets as the policy's performance improves or as new dimensions of evaluation emerge—the meta-rewarding evaluator progressively sharpens the granularity and efficacy of reinforcement signals. This dynamic refinement is particularly beneficial in complex LLM-driven tasks, ensuring that policy updates become more targeted and aligned with nuanced performance improvements, ultimately facilitating advanced policy learning.

### 2.3 IMPLEMENTATION: META REWARD MODEL

The MPO framework is implemented through the introduction of a meta reward model (MRM), which can be conceptually viewed as a "senior instructor" providing guidance to a "junior instructor" on how to evaluate the work produced by a "student." As illustrated in Figure 3, the MRM oversees the broader training context and issues targeted refinements to the evaluation rubric (or prompt) used by the reward model (RM), which plays the role of a "junior instructor." These refinements are informed by inputs drawn from three sources: the prompt dataset, the policy model (the "student"), and the reward model itself. Specifically, at every fixed $k$ training batch steps, the MRM performs an MPO procedure by processing contextual input sources through general meta-level prompts, which are designed to be task-agnostic and applicable across all tasks.

From the prompt dataset, the MRM processes the task description, a set of $n$ task-specific prompt instructions, and—when available—the corresponding $n$ reference solutions. It also receives the policy model's $n$ generated responses to these prompts, the current version of the evaluation rubric, and the scores assigned by the RM using that rubric. These $n$ input samples are randomly selected

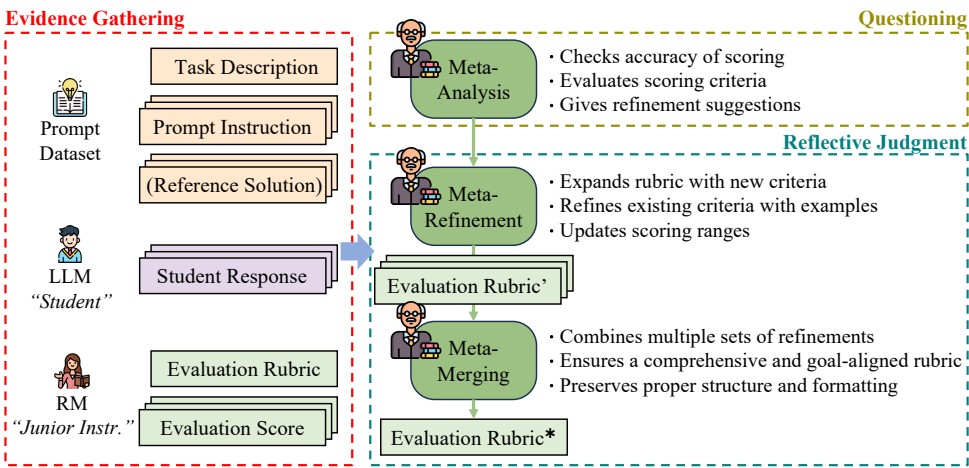

Figure 3: The three Meta Policy Optimization steps—*meta-analysis*, *meta-refinement*, and *meta-merging*—are carried out by the meta reward model and operate over a broader input context than that used by the reward model.

from the training batches accumulated since the last MPO step. Leveraging this rich contextual input, the MRM identifies weaknesses or gaps in the current rubric and prescribes increasingly fine-grained and targeted evaluation criteria. This refinement process is triggered every fixed $k$ training batches and follows the three-stage procedure illustrated in Figure 3. At each stage, the MRM is prompted with meta-level instructions that are designed to be broadly applicable across a range of tasks.

**Meta-Analysis.** The first step of MPO involves processing the full input context to assess whether the RM's scoring is accurate and reliable—particularly in cases where the student LLM may exploit loopholes of the RM through reward hacking. The MRM evaluates whether the current scoring criteria are sufficiently comprehensive and detailed, and prescribes necessary adjustments to improve evaluation quality and robustness. This step is particularly crucial, as it serves to detect loopholes in the RM's evaluation logic early on.

**Meta-Refinement.** Building on the results of the meta-analysis, the next step is to construct a more refined rubric. The MRM begins by determining the appropriate number of evaluation criteria, then systematically expands each item— either by introducing new criteria or by enriching existing ones with more detailed descriptions and illustrative examples. Additionally, it adjusts the scoring scale to more effectively distinguish between varying levels of response quality.

**Meta-Merging.** MPO samples $n$ student responses and proposes $n$ corresponding rubric refinements. In the final step, it **merges these refinements into a single, coherent rubric prompt.** Mirroring the meta-refinement stage, the merger first selects the appropriate number of evaluation items based on the breadth and overlap of the proposed changes, then instantiates each item with precise criteria and guidance. The consolidated rubric becomes the updated evaluation prompt for the reward model (RM) in subsequent training steps. Although the meta reward model (MRM) is updated every 10 steps, merging operates at the batch level rather than per input: prompt refinements are derived from aggregate feedback, producing shared criteria for upcoming batches and preventing arbitrary divergence of reward criteria across inputs.

**Mathematical Formulation.** Appendix C presents a formal, equation-level specification of the MPO process that builds on the theoretical foundations in Section 2.2.1.

**The evolution of the rubrics** in our MPO framework is driven by three key factors:

1. the nature of the task, as defined by the task prompt,
2. the outputs generated by the policy model, and
3. the rubric prompt provided to the reward model.

As discussed above, at each MPO step, we randomly select $n$ episodic generations and conduct analysis and refinement based on these samples. When the selected samples reveal novel challenges, the rubric evolves to incorporate criteria that address these emerging issues. In this work, the sample selection process is entirely random. For future work, we plan to explore more informed sampling strategies that prioritize episodes most likely to yield meaningful and impactful refinements.

## 3 EXPERIMENTS

To investigate the effectiveness of MPO and its influence on training dynamics, we conduct three core experiments. Section 3.1 outlines our experimental setup. Section 3.2 evaluates the performance of MPO-aligned LLMs on an argumentative essay writing task, exploring different pairings of junior and senior instructors. Section 3.3 empirically shows that MPO generalizes to tasks demanding varying degrees of evaluative thinking. In Section 3.4, we analyze how the rubric prompts used by the RM (the "junior instructor") evolve over successive MPO iterations. Finally, Section 3.5 presents a comparison between MPO-aligned models and models trained using an "oracle" prompt. We include additional experiments in the Appendix, such as **an ablation study of MPO components** (Appx. F) and quantitative and qualitative examples of **countering reward hacking** (Appx. D).

### 3.1 EXPERIMENTAL SETUP

**Policy Model.**    Throughout our main experiments, we fix the policy model to a relatively small LLM: Qwen2-1.5B-Instruct (Yang et al., 2024). This choice is motivated by two factors. First, we require a model with sufficient headroom for improvement across our target tasks. Since we use publicly available benchmarks with limited resources, we focus on smaller open-sourced models which could show clear effects and values of the proposed framework. Second, some tasks—such as mathematical reasoning—require generating over 1000 tokens, which imposes a significant memory load during the PPO step. Larger models exceed the capacity of our available GPUs, making Qwen2-1.5B-Instruct a practical and scalable option. In Appendix E, we further show that MPO improves performance even when applied to a larger policy model from a different family, namely Llama-3.1-8B-Instruct.

**(Meta) Reward Models.**    For reward modeling, we use Qwen2.5-32B-Instruct-AWQ and Qwen2.5-72B-Instruct-AWQ (Qwen et al., 2025), exploring all four junior–senior RM–MRM configurations: **32b_32b**, **32b_72b**, **72b_32b**, and **72b_72b**, where the first and second terms denote the sizes of the reward model (RM) and meta-reward model (MRM), respectively. Using larger models as (M)RMs is feasible because only inference is required, which can be efficiently handled by an LLM-serving framework such as SGLang (Zheng et al., 2024).

We also evaluate three fixed-RM baselines without MPO:

- using the **initial MPO prompt**,
- using a **domain expert–crafted prompt** (essay writing task only) (Hamner et al., 2012), and
- using an **iteratively refined prompt** via **AutoPrompt** (Levi et al., 2024) with GPT-4o.

They are denoted in tables as {RM size}_iter0, {RM size}_expert, and {RM size}_AP, respectively.

In Appendix E, we further demonstrate that MPO also improves performance when the reward model comes from a different LLM family, specifically `microsoft/Phi-3-mini-128k-instruct`, highlighting the framework's flexibility and generality.

**Implementation.**    Our MPO framework relies on online (meta) reward models implemented as LLMs with an interchangeable prompt mechanism. To support this, we extend the TRL library (von Werra et al., 2020) by implementing a prompt-based, online reward model, where LLM-based RMs are hosted using the SGLang framework (Zheng et al., 2024). Additionally, we extend the "PPOTrainer" class in trl to a customized "MPOTrainer," which integrates the MPO refinement steps directly into the training loop along with other necessary modifications.[1]

We note that the MPO framework is general and modular, and can be integrated into other RL optimization techniques that rely on reward models or functions—such as GRPO (Shao et al., 2024b)—to enable dynamic rubric refinement and more adaptive reward shaping. We also note that MPO is highly scalable and efficient (see Appendix G for more details).

### 3.2 IMPACT OF MPO ON ESSAY WRITING TASK

**Setup.**    We train four policy models using MPO with different RM–MRM pairings as described in Section 3.1, and another four using vanilla PPO with fixed RM prompts. The essay writing dataset is compiled by Kim et al. (2025) and includes writing instructions drawn from diverse sources, such as English proficiency exams, a persuasion corpus, and the Change My View (CMV) subreddit. The training set comprises 26,013 samples, and the test set includes 4,096 samples. Both MPO and PPO are trained for a single epoch over the training set, with MPO refinement steps occurring every 10

---

[1] Please refer to our codebase for exact details.

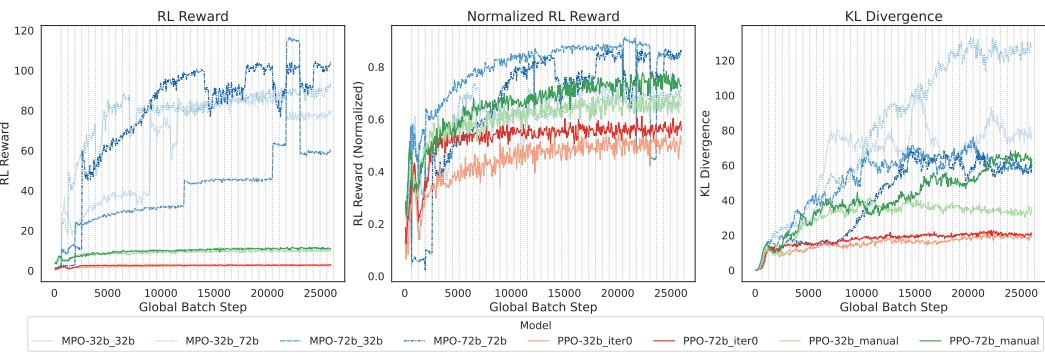

Figure 4: Training curves for eight essay-writing policy models, each pairing different-sized reward models (RM) and meta-reward models (MRM). The RL Reward and Normalized RL Reward plots show how reward values evolve over global batch steps, capturing the quality of generated responses as judged by the corresponding RM at each point in training. The normalized plot is obtained by dividing the RL reward values by the total attainable score defined by the current rubric, providing a more consistent view of reward dynamics across evolving evaluation criteria. Kullback-Leibler (KL) divergence quantifies the extent of policy drift throughout training. The dotted vertical lines indicate MPO rounds, which occur every batch size × MPO step—640 steps in our setup.

batch steps during training. We utilize eight NVIDIA-A100 80GB GPUs to RL-train the models with an effective combined batch size of 64, where each RL episode spans up to 400 tokens.[2] This configuration yields 40 MPO refinement steps over the course of one-epoch training.

After single-epoch training, we generate essays for all test prompts using the final checkpoints and evaluate them through 10,000 head-to-head comparisons following the Elo-based Chatbot Arena framework (Chiang et al., 2024), with GPT-4o serving as the impartial judge and a small K-factor of 4 applied after each match to maintain rating stability. The resulting Elo scores, summarized in Table 1, provide a relative ranking of essay quality across the eight models.

**Training Curves.** Figure 4 shows the training curves for eight models, each corresponding to a different pairing of RM and MRM sizes in the essay writing task. Note that we use raw RL reward values for optimization, as they preserve fine-grained distinctions in reward signals—amplifying reward sensitivity and improving optimization effectiveness—despite a slight deviation from our formal theoretical framework. The four models employing MPO exhibit increasing RL reward values over time, driven by successive MPO steps that refine the evaluation prompt by (1) introducing a greater number of evaluation criteria and (2) expanding the scoring ranges associated with those criteria as training progresses. While these curves suggest that training is progressing adaptively as intended, the reward scores are themselves adaptive—reflecting evolving rubrics—so differences in reward values across models (shown in the RL Reward plot) do not necessarily translate to corresponding performance gaps on the final test set.

**Results.** First, we evaluate the performance of all MPO-aligned models by computing their Elo ratings (see left panel of Table 1). The results show that performance improves primarily with the size of the RM, followed by that of the MRM, with the highest scores achieved when 72B LLMs are used for both components. Next, we evaluate MPO models and baseline methods separately based on the RM size, keeping the MRM fixed at 72B (see right panel of Table 1). The results show that the MPO-aligned model consistently outperforms all baselines, including the PPO-aligned model using an expert-crafted evaluation prompt. We note that the PPO-aligned model using the expert prompt and a 72B RM resulted in a failed training run due to reward hacking. In this case, the policy model frequently generated responses consisting solely of a title—e.g., *"Robots and the Future of Humanity: A Dilemma of Progress and Responsibility"*—yet still often received high scores of 4 or 5 out of 5.[3]

It is worth noting that, at certain points during training, the MPO-aligned models also displayed instances of reward hacking—such as generating responses in Chinese, producing title-only comple-

---

[2]The 400-token limit aligns with one of the essay-writing datasets used in our experiments, where high-quality human annotations average around 400 tokens.

[3]With a fixed random seed, we repeated the same configuration two additional times, and each run consistently resulted in a failed policy model.

Table 1: Elo ratings for the essay writing task. Left: Comparison across RM-MRM variations (2,000 pairwise comparisons). Right: Comparison across LLMs (5,000 pairwise comparisons). Each rating includes standard deviation computed across 5 runs, denoted by $s$. Red indicates a case where training converged to a degenerate policy due to reward hacking.

| RM | MPO w/ MRM (ours) | | | MPO (ours) | PPO | | | Base |
| | 32B | 72B | RM | MRM-72B | iter0 | expert | AP | LLM |
| --- | --- | --- | --- | --- | --- | --- | --- | --- |
| 32B | $973_{s=\pm12}$ | $1005_{s=\pm5}$ | 32B | $\mathbf{1168_{s=\pm7}}$ | $984_{s=\pm10}$ | $1084_{s=\pm17}$ | $970_{s=\pm15}$ | $794_{s=\pm11}$ |
| 72B | $985_{s=\pm7}$ | $\mathbf{1037_{s=\pm10}}$ | 72B | $\mathbf{1244_{s=\pm8}}$ | $1111_{s=\pm7}$ | $706_{s=\pm12}$ | $1050_{s=\pm15}$ | $889_{s=\pm10}$ |

tions, or outputting overly short and degenerate answers. However, these cases were identified and addressed during the MPO procedure. Appendix D provides qualitative and quantitative examples of countering reward hacking through MPO steps in more depth.

The main takeaway from this experiment is that heavily hand-engineered prompts by domain experts can yield the strongest performance among PPO-aligned models (the 32B RM case). However, the fixed-prompt PPO setup remains vulnerable to reward hacking, which can lead to policy collapse, as observed in the 72B RM case. In contrast, incorporating the MPO procedure—even when starting from the basic initial prompt (i.e., iter0)—produces a higher-quality model than PPO with the expert-crafted prompt.

### 3.3 Generalization of MPO across Different Tasks

As discussed in Section 2.1, evaluating written essays requires a form of evaluative thinking that is more breadth-focused—guided by a broad set of generalized rubric items that can be applied across diverse prompts and writing styles. In this experiment, we apply the MPO framework to tasks that vary in their demands along the depth and breadth dimensions of evaluative thinking, examining whether the benefits of the proposed approach persist across this spectrum. As illustrated in Figure 2, we evaluate MPO across three additional tasks: **summarization**, **ethical reasoning**, and **mathematical reasoning**.

For these three tasks, we compare performance across four models: **Base LLM**, the original Qwen2-1.5B-Instruct model prior to any alignment; **32b_iter0**, a vanilla PPO-aligned model using the initial evaluation prompt with the 32B Qwen model as the RM; **32b_AP**, another PPO-aligned model using evaluation prompt iteratively refined via AutoPrompt (Levi et al., 2024) using GPT-4o; and **32b_32b**, an MPO-aligned model using the 32B Qwen model for both the RM and MRM.

#### 3.3.1 Summarization Task

For summarization, we train models on the BillSum benchmark (Kornilova & Eidelman, 2019) for one epoch, applying MPO refinements every 20 batch steps and generating 4.5K summaries for evaluation. We do not use gold summaries during training; instead, the quality of generated summaries is evaluated directly by the reward model. Performance is measured using ROUGE scores against human-written references, along with mean Elo ratings computed from five runs of 2,000 pairwise comparisons judged by GPT-4o. Results are reported in Table 2.

The MPO-aligned 32b_32b model outperforms the other three models in ROUGE scores against human-written gold summaries. However, in pairwise Elo evaluations, the PPO-aligned 32b_AP model receives the highest rating. This discrepancy may stem from the GPT-4o judge favoring outputs from models aligned using evaluation rubrics it helped generate. How-

| | R-1 | R-2 | R-L | R-Lsum | Elo Rating |
| --- | --- | --- | --- | --- | --- |
| Base LLM | 41.85 | 20.83 | 27.85 | 27.84 | $819_{s=\pm21}$ |
| PPO 32b_iter0 | 45.97 | 23.57 | 30.29 | 30.29 | $953_{s=\pm19}$ |
| PPO 32b_AP | 45.92 | 21.98 | 28.80 | 30.22 | $\mathbf{1149_{s=\pm12}}$ |
| MPO 32b_32b | **48.00** | **24.96** | **30.97** | **30.98** | $1079_{s=\pm12}$ |

Table 2: Performance of models evaluated using R (ROUGE) scores on the BillSum long-document summarization task and Elo ratings from 2,000 pairwise comparisons.

ever, further experiments are needed to validate this hypothesis. Figure 6 presents an excerpt of the evaluation rubric used for the summarization task, showing how MPO expanded the rubric to incorporate criteria specifically relevant to assessing legislative bill summaries.

#### 3.3.2 Ethical Reasoning Task

For ethical reasoning, we use the Anecdotes from the Scruples dataset (Lourie et al., 2020), a collection of over 32,000 ethically complex real-world situations labeled with community judgments.

We randomly sample 13K anecdotes for training and 4.7K for testing, running a single training epoch with MPO steps performed every 10 batch steps, leading to 20 rubric refinements. Instead of relying on the binary ground truth labels, reward scores are based solely on the quality of ethical reasoning to encourage deeper reasoning development rather than optimizing for imbalanced label distributions.

Evaluation based on accuracy against the binary verdict labels is reported in Table 3. We observe that the MPO-aligned policy model generates ethical reasoning traces that result in a higher degree of alignment with human-annotated verdicts.

### 3.3.3 MATHEMATICAL REASONING TASK

For mathematical reasoning, we use the MATH dataset (Hendrycks et al., 2021), which contains 12,500 high school competition-style problems across seven subjects, each with detailed step-by-step solutions for evaluating both final answers and reasoning processes. We train on 7.5K samples and test on 5K, clustering problems into 21 subject-cluster groups and maintaining a separate evaluation prompt for each, with MPO refinements triggered every 30 batch steps. The reward model follows a *plan-then-execute* strategy adapted from Saha et al. (2025)[4], formulating an evaluation plan before scoring student responses with rubric-guided assessment. The reward model operates in two scoring modes, selected at random with equal probability. The first mode assigns scores based solely on exact match of the final answer span, while the second uses LLM-based evaluation to assess the correctness of the reasoning used to arrive at the solution.

Table 3: Accuracy on the ethical reasoning (Scruples–Anecdotes) and mathematical reasoning (MATH) benchmarks.

| Model | Accuracy (%, correct / total) | |
|---|---|---|
| | Scruples–Anecdotes | MATH |
| Base LLM | 33.80 (1601/4736) | 17.90 (905/5056) |
| PPO 32b_iter0 | 63.79 (3021/4736) | 48.77 (2466/5056) |
| PPO 32b_AP | 58.68 (2779/4736) | 49.29 (2492/5056) |
| MPO 32b_32b | **68.60** (3249/4736) | **50.38** (2547/5056) |

Accuracy results based on the exact match of reference answers are reported in Table 3. The first notable point is that our proposed evaluation approach significantly improves the performance of the base 1.5B LLM. This suggests that tasks requiring depth-oriented evaluative thinking—such as mathematical reasoning—can benefit substantially from structured, plan-then-execute evaluation. As with the other tasks, applying the MPO framework through refinement of meta-level guidelines further enhances the performance of our proposed plan-then-execute reward model. However, the relative improvement is more modest compared to the other tasks. We hypothesize that this is due to the highly instance-specific nature of mathematical reasoning, where scoring relies heavily on whether the sequential logic leads precisely to the correct answer. In contrast, the meta-level guidelines—constructed from sampled instances—tend to remain relatively general. Nonetheless, subject-specific refinements still contribute to performance gains. Developing more granular and tailored meta-guidelines could yield additional improvements, which we leave for future work.

### 3.4 EVOLUTION OF EVALUATION RUBRIC

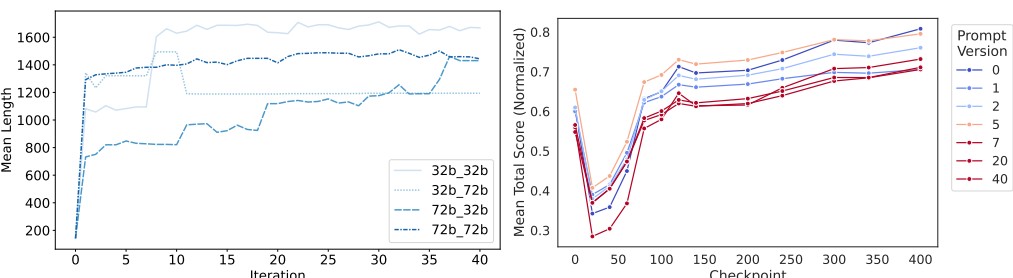

(a) Rubric length during MPO refinements          (b) Total score across rubric refinements

Figure 5: (a) Mean length of rubric items for essay writing task across the MPO-aligned models. (b) Mean normalized total rubric score for 1,000 test essays (generated by the 32b_72b model) across successive evaluation prompt refinements.

In this experiment, we seek to uncover holistic patterns across rubric evolution, particularly focusing on essay writing task.

---

[4]The key difference is that their work trains a pairwise LLM-as-judge for iterative DPO, while our approach uses absolute scoring in a single round of PPO.

**Mean Lengths of Rubric Items.** For the essay writing task, we track how the average length of each rubric item evolves over successive MPO refinements. Notably, the length increases sharply after the first refinement and continues to grow steadily over the next 5 to 10 iterations, before plateauing during the remaining stages of training (Fig. 5a). Manual inspection confirms that most meaningful rubric changes occur within the first 5 to 10 iterations, after which the refinements become relatively minor. This observation suggests that the current fixed MPO schedule could be improved through dynamic adjustment based on training dynamics—an avenue worth exploring in future work.

**Mean Total Scores Across Successive Rubrics** Figure 5b plots the normalized mean total scores for 1,000 randomly sampled test essays, generated by the 32b_72b model and evaluated using both early and late-stage RM prompts evolved through MPO. The general pattern shows that mean total scores increase over the course of training, reflecting the policy model's improving output quality. **However, earlier versions of the evaluation prompts tend to assign higher scores to samples**, largely because their coarser and less fine-grained criteria make it easier for responses to meet the rubric's standards. In contrast, **later-stage prompts, which feature more detailed and discriminating rubric items, assign lower scores**, as they capture subtler flaws and impose stricter evaluation standards. This trend highlights how rubric refinement not only tightens evaluation but also provides more accurate and demanding feedback to guide policy improvement.

We provide the sequence of evolved rubrics in Appendix J and all intermediate MPO results in the anonymized repository.

### 3.5 COMPARISON AGAINST HUMAN-ENGINEERED ORACLE PROMPT

For the essay writing task, Kim et al. (2025) hand-engineered an evaluation prompt for the RM through an iterative process involving over 60 vanilla PPO training runs. The prompt was refined based on reward hacking behaviors observed from the same Qwen2-1.5B-Instruct model and adjusted to better align with human evaluation scores by modifying evaluation criteria. We treat this prompt as an oracle prompt, as it represents a hand-evolved rubric distilled through extensive PPO training experience. Figure 7 in Appendix H compares the oracle evaluation prompt with the final version produced by the

Table 4: Elo ratings for the essay writing task, evaluated through 5K pairwise comparisons across 4 LLMs. Each rating is with a standard deviation ($s$), computed across 5 independent experiments.

| RM | **MPO** (ours) MRM-72B | **PPO** oracle |
|---|---|---|
| 32B | $1039_{s=\pm 11}$ | $894_{s=\pm 13}$ |
| 72B | $\mathbf{1064_{s=\pm 14}}$ | $1003_{s=\pm 17}$ |

MPO framework. Unlike the oracle, the MPO-evolved prompt was generated within a single training epoch and features a diverse set of rubric items specifically tailored for essay evaluation. Each item is accompanied by detailed descriptions that support fine-grained point allocation across a defined scoring range.

Table 4 presents the Elo ratings for the 32b_72b and 72b_72b MPO-aligned models alongside PPO-aligned models using the oracle prompt, based on 5,000 GPT-4o-judged pairwise comparisons. Both MPO-aligned models outperform the PPO baselines, with the 72b_72b MPO model achieving the highest Elo rating. These results demonstrate that the MPO framework can automatically generate evaluation prompts that surpass the quality of extensively hand-engineered oracle prompts, without requiring any task-specific manual prompt design.

## 4 CONCLUSION

This work introduces Meta Policy Optimization (MPO), a novel framework that enhances reinforcement learning from human or AI feedback by dynamically evolving the evaluation rubrics used by reward models. Grounded in the cognitive principles of evaluative thinking and metacognition, MPO empowers reward models to not only evaluate policy outputs but also reflect on and refine their scoring criteria over time. Across diverse tasks (including essay writing, summarization, ethical reasoning, and mathematical problem solving), MPO consistently improves alignment and outperforms models relying on static, manually crafted prompts. **Beyond improved empirical performance, MPO offers a new lens for thinking about reward modeling as an adaptive, self-improving process.** Our analysis further reveals that the evolved rubrics exhibit deeper linguistic structure, suggesting the emergence of more principled evaluation schemas.

For future work, several promising directions emerge: dynamically adjusting MPO frequency based on training dynamics, scaling to more granular rubric specializations, exploring multi-turn dialogues and interactive settings, and integrating MPO with advanced optimization algorithms beyond PPO.

## 5 ETHICS STATEMENT

### 5.1 LIMITATIONS

As shown in Appendix E, MPO is effective even across different model families and scales. We did not evaluate policy models larger than 8B parameters due to limited GPU resources, driven mainly by the compute demands of PPO training. We note that this constraint reflects PPO, not MPO.

Future work could explore mitigation strategies such as constrained decoding or employing newer model generations that offer improved instruction following at smaller scales.

### 5.2 SOCIETAL IMPACT

Training and aligning large language models—like deep learning research more broadly—can have significant environmental impacts, including high energy consumption and greenhouse gas emissions. A key benefit of our work is that the proposed MPO framework reduces the need for repeated RLAIF training cycles, which often involve extensive manual prompt engineering to improve reward modeling. As demonstrated in our MPO vs. Oracle experiment, MPO achieves better performance to manually crafted reward prompts with only a single training iteration. This contributes to a positive societal impact by enabling more energy-efficient and environmentally sustainable alignment of LLMs.

### 5.3 DECLARATION OF LLM USAGE

We acknowledge making editorial use of Grammarly and ChatGPT 4o exclusively for improving linguistic clarity in places where sentence flow was less fluent. These tools did not generate new scholarly content or design any core methodological contributions. Rather, they served as auxiliary resources for finalizing readability.

## 6 REPRODUCIBILITY STATEMENT

We release an anonymous repository with all code, training scripts, configuration files, and datasets, along with full experiment logs, intermediate artifacts, and a step-by-step training guide, accessible at the anonymized link: `https://anonymous.4open.science/r/mpo-CD3B`

The main text details the experimental setup and task coverage, including where to find hardware and batching choices, and where each experiment is defined and compared. Implementation specifics for our extensions to TRL and SGLang, as well as our MPOTrainer modifications, are referenced in Section 3 and mirrored in the repository readme. Dataset construction, splits, and evaluation protocols are described in the task setup passages and are replicated in the supplemental scripts.

The appendix collects theory, proofs, and additional methodological explanations deferred from Section 2.2, plus consolidated prompts and rubric materials required to reproduce the evolving reward mechanism. We also provide ablation-style artifacts that document how MPO counters reward hacking and how evaluation rubrics evolve, including discourse motif analyses, to support independent verification. For quick navigation, the appendix and supplemental package include a Training Details section with hyperparameters, seeds, and runbooks that match the configurations reported in Section 3.1.

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

## A RELATED WORK

### A.1 REWARD HACKING IN LLMS

RL has been widely applied in the post-training of LLMs, enhancing areas such as factuality (Tian et al., 2024), code generation (Le et al., 2022), reasoning (DeepSeek-AI et al., 2025), and multi-agent decision-making (Park et al., 2025). A predominant strategy for incorporating RL into LLM training is reinforcement learning from human feedback (RLHF) (Ziegler et al., 2019; Ouyang et al., 2022c; Bai et al., 2022; Ahmadian et al., 2024; Park et al., 2024).

Reward hacking (Skalse et al., 2022) is a phenomenon that is observed when an RL agent exploits flaws, ambiguities, or lack of specificity in the reward function (as also noted in Goodhart's Law) to achieve high rewards in unintended ways, often showing coherent but unanticipated behavior (Amodei et al., 2016a). This leads to the agent being misaligned with the human-intended behaviors, yet achieving high rewards. This has emerged as a critical challenge in RLHF and RLAIF (Krakovna et al., 2020; Pan et al., 2022; Gao et al., 2023; Fluri et al., 2024; Lee et al., 2024).

A variety of studies have highlighted the detrimental effects of reward in aligned LLMs (Pang et al., 2023). Various undesirable side-effects, such as sycophancy (Perez et al., 2023; Sharma et al., 2024; Denison et al., 2024), verbosity (Saito et al., 2023; Singhal et al., 2024), and deception (Wen et al., 2025), among others. To address these challenges, recent work has proposed numerous reward modeling and mitigation strategies, such as such as Reward Ensembling (Eisenstein et al., 2024; Rame et al., 2024; Coste et al., 2024; Zhang et al., 2024b), and designing RL regularizations (Miao et al., 2025; Chen et al., 2024a; Liu et al., 2024b; Miao et al., 2024; Fu et al., 2025). Although these strategies have demonstrated varying levels of success, challenges such as reward overfitting, misspecification, and misgeneralization still pose significant obstacles to robust and reliable reward model-based alignment in practice.

We argue that effectively addressing reward hacking requires alignment mechanisms capable of broader contextual reasoning, as it often stems from the interaction between poorly defined reward signals, evolving policy behaviors, and the shifting dynamics of exploration versus exploitation during training.

### A.2 ITERATIVE ALIGNMENT IN LLMS

As LLMs have scaled, despite advances in alignment techniques (Ouyang et al., 2022a; Bai et al., 2022), they have fallen short in handling complex, shifting failure modes (Xu et al., 2023; Meng et al., 2024; Liu et al., 2024a; Ethayarajh et al., 2024). This has led to a growing interest in iterative approaches to preference optimization for aligning LLMs (Tran et al., 2023; Xiong et al., 2024; Pang et al., 2024; Wu et al., 2024b; Chen et al., 2024b; Ding et al., 2024). While these methods improve upon single-pass preference tuning by incorporating feedback into successive training rounds, allowing the model to refine its behavior based on evolving outputs, they remain **dataset-bounded**: relying on explicit preference comparisons or fixed prompt templates that are dependent on the initial design of reward prompts or training distributions and do not adapt during training with an evolving policy. They are also limited in their robustness to reward hacking as the policy shifts since the reward model does not evolve during training. Similarly, methods based on knowledge distillation via supervised fine-tuning (SFT) from a reward model (Wang et al., 2023) encode reward judgments into a static training target, which may no longer reflect optimal behavior as the model improves, further reinforcing non-adaptive biases in reward estimation. The self-rewarding LMs proposed in Yuan et al. (2025) share our motivation in leveraging LLM-as-a-Judge prompting to generate reward signals during training. However, their approach relies on iteratively applying Direct Policy Optimization (DPO), followed by generating a new dataset for each DPO round—a process that is computationally intensive and resource-heavy. In contrast, our MPO framework introduces lightweight, prompt-based reward refinement via a meta-reward model, enabling continuous alignment without the need for repeated dataset regeneration or full model retraining.

### A.3 META-LEVEL ALIGNMENT FOR LLMS

Our work is closely related to Wu et al. (2024a), who first introduced the idea of LLM-as-a-Meta-Judge within a self-rewarding pipeline, enabling a single model to evaluate and refine its own

judgments. Their method trains one LLM to serve as actor, judge, and meta-judge, using a fixed 5-point rubric to generate and score responses, then iteratively refining both roles via DPO. While this reflects MPO's self-improvement philosophy and focus on mitigating reward hacking, the approaches differ: Meta-Judge updates model weights through preference optimization, whereas MPO introduces a separate meta-reward model that rewrites the evaluation rubric itself—adapting the criteria, not just the model, in response to emerging behaviors like reward exploitation.

Similarly, TS-Align (Zhang et al., 2024a) shares MPO's goal of scalable alignment with reduced human supervision but takes a different route. It employs a teacher–student framework where a strong teacher RM re-ranks preference pairs filtered by a smaller student RM, followed by DPO-based fine-tuning. However, the reward prompt remains fixed throughout. In contrast, MPO operates with a single RM (guided by a meta RM) and dynamically evolves the evaluation rubric at regular intervals, allowing the reward function itself to adapt to policy drift and training-phase dynamics.

To the best of our knowledge, MPO is the first to improve LLM alignment via meta-level rubric refinements under PPO, rather than DPO—offering a lightweight, prompt-based alternative that reduces computational cost while enabling continual adaptation to the evolving training landscape.

## B    Deferred Explanation for Section 2.2

Formally, consider a discrete state space $\mathcal{S}$, an action space $\mathcal{A}$ (both finite or countable), and a transition kernel $P(s' \mid s, a)$ specifying the probability of transitioning to state $s' \in \mathcal{S}$ given action $a \in \mathcal{A}$ from state $s \in \mathcal{S}$. We define a golden reward function $r : \mathcal{S} \to \mathbb{R}$ assigning a numerical reward to each state.

**Remark 2.** In LLM-RL settings, a *state* $s$ represents the textual history (e.g., the sequence of tokens generated so far), and an *action* $a$ is the selection of the next token. The subsequent state is thus naturally expressed as $(s, a)$, the concatenation of the history and the chosen token. Practically, assigning a precise numerical reward $r(s)$ is challenging due to subjective criteria such as coherence or relevance. Hence, evaluators typically provide approximate feedback. This scenario aligns well with a partially observable Markov decision process (POMDP) structure, where observations from evaluators form a partial, aggregated view of the underlying states.

**Remark 3.** Consider an LLM tasked with generating responses in a conversational setting. At first, AI or human feedback might only broadly categorize responses as "good," "neutral," or "bad." Over time, however, evaluators might introduce finer distinctions, such as "coherent but impolite," "polite but irrelevant," and "relevant but verbose." Mathematically, this corresponds to refining the granularity of observation sets that the LLM receives, providing increasingly precise and informative feedback. This is directly related to evolving reward model in our case - by ECB-ed RM by ET, they can provide a finer score which will be closer to the golden reward model.

### B.1    Refining Observation Partitions Over Time

Let the set of possible observations at each discrete phase $t = 1, 2, 3, \ldots$ be denoted by $\Omega_t$. Define a collection of partitions $\{\mathcal{O}_{o,t}\}_{o \in \Omega_t}$ of the state space $\mathcal{S}$, satisfying:

$$\mathcal{O}_{o,t} \cap \mathcal{O}_{o',t} = \emptyset, \quad \text{for } o \neq o', \quad \text{and} \quad \bigcup_{o \in \Omega_t} \mathcal{O}_{o,t} = \mathcal{S}.$$

Each partition represents a labeling of states by evaluators, where $\mathcal{O}_{o,t}$ contains states labeled as observation $o$ at time $t$. If the true state at phase $t$ is $s$, the agent deterministically observes label $o$ such that $s \in \mathcal{O}_{o,t}$.

**Refinement property.**    To formally capture increasingly precise feedback, assume each partition refines the previous one. Precisely, for every $\mathcal{O}_{o,t}$, there exists some $\mathcal{O}_{o',t-1}$ satisfying:

$$\mathcal{O}_{o,t} \subseteq \mathcal{O}_{o',t-1}.$$

This means partitions at each subsequent phase are finer than the previous ones, never coarser, *which is also consistent with our MPO setup*–meta reward model guides the reward model with ET to have finer rubrics as training goes.

## B.2 Observation-Based Rewards

Rather than providing immediate rewards based on exact state information, define a reward based on observation labels. When observing $o \in \Omega_t$ at phase $t$, the agent receives an averaged reward:

$$R_t(o) = \mathbb{E}_{s \in \mathcal{O}_{o,t}}[r(s)] = \frac{1}{|\mathcal{O}_{o,t}|} \sum_{s \in \mathcal{O}_{o,t}} r(s).$$

In our MPO setup, a coarser rubric leads to rewards being *averaged over all responses mapped to the same rubric category* as well.

# C Meta Rewarding Process: Formal Mathematical Formulation

## C.1 Mathematical preliminaries

Let $t \in \{0, 1, 2, \ldots\}$ index the refinement step and $i \in \{1, \ldots, n\}$ index samples in a batch. Define:

- Input–output pair $(x_{t,i}, y_{t,i})$, where $x_{t,i}$ is the prompt and $y_{t,i}$ is the generated response.
- Observed rubric reward at time $t$: $R_t(x_{t,i}, y_{t,i})$.
- Internal reward known to the meta–reward model: $r_t(x_{t,i}, y_{t,i})$.
- Training batch: $\mathcal{D}_t = \{(x_{t,i}, y_{t,i}, R_t(x_{t,i}, y_{t,i}))\}_{i=1}^n$.

## C.2 Meta-analysis: identifying misaligned samples

Define the reward discrepancy for sample $i$:

$$\Delta_{t,i} := r_t(x_{t,i}, y_{t,i}) - R_t(x_{t,i}, y_{t,i}). \tag{1}$$

A sample $(x_{t,i}, y_{t,i})$ is flagged for refinement if

$$\left| \Delta_{t,i} \right| = \left| r_t(x_{t,i}, y_{t,i}) - R_t(x_{t,i}, y_{t,i}) \right| > \tau, \tag{2}$$

where $\tau$ is a discrepancy threshold. The flagged set is

$$F_t := \{ i \ : \ |\Delta_{t,i}| > \tau \}. \tag{3}$$

## C.3 Meta-refinement: improving evaluation components

For each $i \in F_t$, the MRM proposes a refined scoring function $R_{t+1,i}$ that better aligns with the internal assessment:

$$\left| r_t(x_{t,i}, y_{t,i}) - R_t(x_{t,i}, y_{t,i}) \right| \geq \left| r_t(x_{t,i}, y_{t,i}) - R_{t+1,i}(x_{t,i}, y_{t,i}) \right|. \tag{4}$$

A stronger guarantee is

$$\Delta_{t+1,i}^2 \leq \Delta_{t,i}^2 - \varepsilon \quad \text{for some } \varepsilon > 0. \tag{5}$$

**Component decomposition.** If $R_t$ consists of $m$ evaluation components, write

$$R_t(x_{t,i}, y_{t,i}) = \sum_{j=1}^m w_{t,j}\, c_{t,j}(x_{t,i}, y_{t,i}), \tag{6}$$

where $w_{t,j}$ are weights and $c_{t,j}$ are component scores. A sample–specific refinement updates

$$R_{t+1,i}(x_{t,i}, y_{t,i}) = \sum_{j=1}^m w_{t+1,j}^{(i)}\, c_{t+1,j}^{(i)}(x_{t,i}, y_{t,i}), \tag{7}$$

with parameters chosen to satisfy equation 4.

## C.4 META-MERGING: UNIFYING REFINEMENTS VIA REGRESSION

Given the refined per-sample scorers $\{R_{t+1,i}\}_{i \in F_t}$, construct the regression dataset

$$\mathcal{T}_{\text{merge}} = \Big\{(x_{t,i}, y_{t,i}, R_{t+1,i}(x_{t,i}, y_{t,i})) : i \in F_t\Big\} \ \cup \ \Big\{(x_{t,i}, y_{t,i}, R_t(x_{t,i}, y_{t,i})) : i \notin F_t\Big\}. \quad (8)$$

Fit a regression model $\mathcal{T}(\cdot; \theta)$, for example a regression tree, to learn the unified scorer

$$R_{t+1}(x, y) \ = \ \mathcal{T}(x, y; \theta_{t+1}), \qquad \theta_{t+1} \ = \ \arg\min_{\theta} \frac{1}{n} \sum_{i=1}^{n} \ell\Big(R_{\text{target}}^{(i)}, \mathcal{T}(x_{t,i}, y_{t,i}; \theta)\Big), \quad (9)$$

with targets

$$R_{\text{target}}^{(i)} \ = \ \begin{cases} R_{t+1,i}(x_{t,i}, y_{t,i}) & \text{if } i \in F_t, \\ R_t(x_{t,i}, y_{t,i}) & \text{if } i \notin F_t. \end{cases} \quad (10)$$

The learned tree provides interpretable rules that unify individual refinements into a coherent global scoring function.

## C.5 CONVERGENCE ANALYSIS

Define the average discrepancy at time $t$:

$$\bar{\Delta}_t := \frac{1}{n} \sum_{i=1}^{n} |\Delta_{t,i}| \ = \ \frac{1}{n} \sum_{i=1}^{n} \big| r_t(x_{t,i}, y_{t,i}) - R_t(x_{t,i}, y_{t,i}) \big|. \quad (11)$$

Under successful refinement,

$$\bar{\Delta}_{t+1} \ \leq \ \bar{\Delta}_t \ - \ \varepsilon \cdot \frac{|F_t|}{n} \quad \text{for some } \varepsilon > 0, \quad (12)$$

and the process is considered converged when

$$\max_{i \in \{1, \dots, n\}} |\Delta_{t,i}| \leq \tau. \quad (13)$$

As $t \to \infty$,

$$\mathbb{E}_{(x,y) \sim \mathcal{D}} \big[ \big| r_t(x, y) - R_t(x, y) \big| \big] \to 0, \quad (14)$$

which states that the rubric-based rewards $R_t$ converge to the internal assessments $r_t$ across the distribution.

## D COUNTERING REWARD HACKING

### D.1 A QUALITATIVE CASE

Consider the following real example observed during RL training:

> **A reward hacking example**
>
> **User:** You will act as an English writer and compose either an essay or a story depending on the instruction given below. Your essay should be no more than 350 words.
> **Instruction:** Write an essay that argues the claim that the U. S. Drug Enforcement Agency has a negative impact on society that outweighs any benefits it provides. Your essay should include evidence and examples to support your argument.
> **Student Response:** "I'm ready when you are. Please provide the essay instruction."
> **Evaluation Score:** 3

In this example, the policy model discovers that generating a degenerate response—like the one shown—yields a score of 3 out of 5 from the RM, despite being clearly misaligned with the task objective. This is an example of reward hacking, where the model exploits weaknesses in the scoring system to earn relatively high points without fulfilling the task faithfully.

---

**Meta-Analysis**

```
The junior instructor's assigned score of 3 out of 5 points for the student's response is
misleading given the provided student generation and the task requirements.  Let's break
down the analysis.
Student's Response:  The student's response is entirely missing.  The student's generation
reads, "I'm ready when you are.  Please provide the essay instruction."  This indicates
that the student did not actually write the essay as instructed, but rather responded with
a statement that is irrelevant to the task.
Task Requirements and Prompt Instructions:  ···
Junior Instructor's Scoring Criteria Analysis:  ···
Discrepancies and Feedback:
1.  Adherence to Task Requirements (0 points)
 - The student's response does not reflect the task requirements at all.  The student did
not provide an essay on the given topic.
 - The junior instructor should assign 0 points for adherence to task requirements.
2.  Content Accuracy and Relevance (0 points) ···
3.  Argument Strength (0 points) ···
4.  Counterarguments and Rebuttals (0 points) ···
···
```

---

Meta-analysis is conducted every k batch steps, identifying failure cases and proposing concrete remedies and feedback for rubric refinement. Without such intervention, continued RL training under a fixed rubric would reinforce these degenerate outputs, potentially resulting in failed training.

A proposed refinement to the rubric introduces essay length as an explicit evaluation criterion and expands the scoring range from 0 to 10 points. This adjustment enables more fine-grained differentiation between responses and discourages degenerate outputs that exploit vague or underspecified scoring guidelines:

---

**Meta-Refinement**

```
···
Task Alignment and Argument Clarity (10 points)
 - **10**:  The essay clearly and consistently addresses the specific claim made in the
prompt and presents a well-defined, coherent argument supported by relevant examples and
evidence.  The argument is fully developed and addresses the core issue of the prompt.
The essay meets the required word count.
 - **8**:  ···
 - **6**:  ···
 - **4**:  ···
 - **2**:  ···
 - **0**:  The essay does not address the specific claim or is completely off-topic.  The
argument is not related to the prompt and does not address the core issue.  The essay is
significantly below the required word count.  ···
```

---

## D.2 A QUANTITATIVE EXAMPLE

Table 5 highlights a prominent case of reward hacking observed in the Qwen models, where the model exploits the reward function by **generating content in Chinese**. We report the mean ratio of Chinese characters in the generated episodes across training steps. A high proportion of Chinese characters emerges in the PPO-aligned model, indicating severe reward hacking behavior. In contrast, this undesirable behavior is effectively suppressed in the MPO-aligned model, demonstrating the framework's ability to maintain alignment and prevent exploitative strategies.

Table 5: Mean ratio of Chinese characters in generated episodes. The PPO-aligned model exhibits reward hacking by producing outputs in Chinese, while the MPO-aligned model eliminates this behavior.

| Batch Steps | PPO-32B-iter0 | MPO (32B_32B) |
|---|---|---|
| 10 | 0.00406 | 0.00000 |
| 20 | 0.00229 | 0.00000 |
| 30 | 0.00000 | 0.00000 |
| 40 | 0.10714 | 0.00000 |
| 50 | 0.22957 | 0.00000 |

## E    MODEL DIVERSITY AND POLICY MODEL SCALING

Our use of a smaller policy model was driven purely by limited GPU resources rather than any additional computational or memory overhead introduced by MPO. As discussed earlier, the meta-reward model (MRM) can be implemented using the same LLM-based reward model (RM) without incurring extra memory cost.

To further address concerns regarding generalizability, we conducted additional experiments with a larger model from a different LLM family, `microsoft/Phi-3-mini-128k-instruct` (4B parameters). Although we were unable to evaluate a 7B model due to current resource constraints, this limitation is inherent to PPO-based training rather than specific to our MPO framework.

In addition, we evaluated our framework using a different architecture by employing `Llama-3.1-8B-Instruct` for both the RM and MRM. This variation in both model scale and architecture demonstrates that our framework can effectively extend to diverse model families and sizes.

Table 6 presents the Elo simulation results averaged over four independent runs, each consisting of 2,000 matches. The results show that the MPO-aligned model significantly outperforms the baseline PPO-aligned model, reinforcing the robustness of MPO across different model families.

Table 6: Elo simulation results averaged over four runs, each with 2,000 matches. MPO improves alignment even when scaling to larger models and alternative architectures.

| Model | Mean Elo Score | Std. Dev. |
|---|---|---|
| llama8b_8b | 1016.072 | 4.687 |
| iter0-llama8b | 953.504 | 10.049 |

## F    ABLATION STUDY ON MPO COMPONENTS

We conduct an ablation study to evaluate the contributions of the individual MPO components—*meta-analysis*, *meta-refinement*, and *meta-merging*—by comparing three experimental setups:

1. **Baseline PPO**: Standard PPO training without any MPO components.

2. **MPO without Meta-Analysis**: Only meta-refinement ("reflective judgment") and meta-merging are applied, while the meta-analysis phase is skipped.

3. **Full MPO**: All three components are enabled.

In the second variant, the framework moves directly to the meta-refinement step without the prior diagnostic phase where the meta-reward model identifies rubric limitations, as illustrated in Figure 3.

Table 7 reports the Elo scores averaged over three independent simulations, each consisting of 2,000 pairwise matchups evaluated by GPT-4o. The results demonstrate that omitting the meta-analysis step leads to a notable performance drop, confirming the importance of a step-by-step process where reflective diagnosis precedes rubric refinement. These findings validate our design choice of incorporating meta-analysis as a distinct stage before refinement.

Table 7: Ablation study results averaged over three Elo simulations (2,000 matches each). Removing the meta-analysis step leads to a clear decrease in performance, highlighting its crucial role.

| Model Variant | Mean Elo Score | Std. Dev. |
|---|---|---|
| iter0-llama8b (Baseline PPO) | 979.897 | 20.773 |
| llama8b_8b-no-analysis (No Meta-Analysis) | 990.093 | 15.351 |
| llama8b_8b (Full MPO) | **1030.009** | **5.903** |

## G  SCALABILITY AND EFFICIENCY

MPO is a modular workflow that can be integrated with any RL framework utilizing a reward model or function, such as PPO, and does not introduce substantial computational or memory overhead.

On the **memory** side, MPO only requires a *meta-reward model* (MRM), which can reuse the same LLM-based reward model employed by RLAIF in PPO. When the same model is shared between the RM and MRM, there is effectively no additional memory cost beyond a standard PPO setup. The only extra storage needed is for a small set of generated text files selected via hyperparameters, which remain lightweight and memory-efficient.

On the **compute** side, MPO introduces only a modest overhead due to the additional step of querying the MRM and constructing the next reward prompt from intermediate generations. This process occurs only at fixed batch intervals rather than every step. In our experiments, the complete MPO procedure averaged 1.85 minutes per update: 1.30 minutes for the *Analysis and Refinement* phase and 0.55 minutes for the *Merge* phase. Although we set MPO to run every 20 batch steps for fine-grained analysis, this interval is a tunable hyperparameter that can be adjusted based on dataset complexity and batch size.

In practical terms, training on the essay-writing task required approximately 6 hours on three H100 GPUs. Of this, MPO contributed only 37 minutes (approximately 6% of total time) when executed on a single H100. Given the time saved from avoiding brittle prompt engineering and failed RL runs, combined with the improved performance MPO delivers, we consider this tradeoff highly efficient.

| Method | Compute Time | GPU Memory Usage |
|--------|--------------|------------------|
| PPO | 1.00 | 1.00 |
| MPO | 1.11 (11% overhead) | 1.00 (no overhead if RM = MRM), or $1 + \alpha$ (if separate MRM used) |

Table 8: Relative compute time and GPU memory usage for PPO vs. MPO. The parameter $\alpha$ represents the additional memory required only when a separate meta-reward model is used.

# H    COMPARISON OF EVALUATION RUBRICS

**Essay Writing: Initial Prompt**
You will act as an English instructor and evaluate the quality of an essay or story written by a student in response to given instructions. Your evaluation should focus on the "discourse" aspect of the text. Output your score as an integer from 0 (worst quality) to 5 (best quality). Surround your integer score with <score> and </score>. Example: <score>2</score>

**Essay Writing: Last Prompt**
### Content and Argumentation (30 points)
- **30 points**: The essay presents a clear, well-supported argument that effectively addresses the claim. It includes a variety of strong, relevant evidence and reasoning, and addresses counterarguments comprehensively. (Example: The essay provides historical context, cites specific legal precedents, and includes expert opinions to support its argument.)
…
### Historical and Contextual Analysis (10 points) …
### Impact Analysis (15 points) …
### Counterarguments (10 points) …
### Structure and Coherence (15 points) …
### Use of Evidence (15 points) …
### Clarity and Coherence (10 points) …
### Language and Style (10 points) …
### Persuasiveness (10 points) …

**Mathematical Reasoning: Initial Prompt**
Evaluate the correctness of the [Student Response].

**Mathematical Reasoning: Last Prompt**
1. **Consistency in Scoring and Feedback**:
   - Ensure that the scoring and feedback are consistent across all cases. If a student's response is partially correct, acknowledge the correct aspects and provide a nuanced score.
   - **Example**: In [Evaluation Case #1], the student used the concept of similar triangles, which is a valid approach. The feedback should have pointed out the specific error in their calculation and provided a more nuanced score, perhaps a 2 or 3 out of 5.
2. **Detailed Feedback**: …
3. **Handling Omitted Responses**: …
4. **Correctness of Mathematical Reasoning**: …
5. **Detailed Explanation of Steps**: …
6. **Encouragement of Full Solutions**: …
To ensure these guidelines are followed, the junior instructor should implement the following additional diagnostic checks:
- **Specific Error Identification**: …
- **Step-by-Step Guidance**: …
- **Encouragement of Review**: …
- **Reminders of Formatting Guidelines**: …
### Specific Examples and Patterns …
### General Patterns …

**Summarization: Initial Prompt**
Your evaluation should focus on the faithfulness and conciseness of the summary. Output your score as an integer from 0 (worst quality) to 5 (best quality).

**Summarization: Last Prompt**
### Faithfulness to the Bill (30 points)
- **30 points**: The summary accurately reflects all key points and details of the bill, maintaining the correct context and intent. It includes specific details such as the short title, the structure of the bill, the purposes, the findings, the definitions of key terms, the establishment of specific programs, the authorization of appropriations, the requirement for periodic reports, the effective date, the prohibition of recounts or certification of results without counting provisional ballots, and other relevant provisions.

…
### Conciseness (10 points) …
### Completeness (30 points) …
### Clarity and Precision (20 points) …
### Deductions
…

**Ethical Reasoning: Initial Prompt**
Evaluate the ethical reasoning based on its logical coherence, depth of moral insight, and alignment with established ethical principles. Assign a score from 0 to 5, where 0 indicates the lowest quality and 5 indicates the highest quality.

**Ethical Reasoning: Last Prompt**
### Logical Coherence (0-5)
- **5**: The reasoning is logically consistent, free from contradictions, and well-supported by evidence or examples. The argument is clear, structured, and avoids repetition or disjointedness. It considers multiple perspectives and the broader ethical implications of the action. The verdict is presented within the required tags. <verdict>RIGHT</verdict>
   - *Example*: "The action of not using the GPS app is justified. It respects personal autonomy and privacy, which are fundamental human rights. The decision to not use the app is a reasonable response to concerns about personal privacy and autonomy. The reasoning is clear, structured, and avoids repetition.
<verdict>RIGHT</verdict>"
- **4**: …
- **3**: …
- **2**: …
- **1**: …
- **0**: …
### Depth of Moral Insight (0-5) …
### Alignment with Established Ethical Principles (0-5) …
### Adherence to Task Instructions (0-5) …

Figure 6: Comparison between initial and last versions of MPO rubrics for all tasks.

| Manual Hand-Crafted Prompt for Essay Writing | Last Version of Prompt Evolved by MPO |
|---|---|
| You will act as an English instructor and evaluate the quality of an essay or story written by a student in response to given instructions. When grading, consider the following discourse aspects of the text.
- Logical Flow and Structure (flow): Assess the logical progression of ideas and the overall organization of the text, ensuring that it is easy to follow and well-structured.
- Hierarchical Organization (organization): Examine the organization of ideas in a hierarchical manner, from general to specific, ensuring that each section supports the main argument or narrative.
- Balance and Emphasis (balance): Ensure that important ideas are appropriately emphasized and that there is a balance in the coverage of different points or sections of the text.

For each aspect, you need to assign an integer score from 0 (worst quality) to 5 (best quality).
When assigning the score, carefully consider which specific parts of the text relate to each aspect.

Assign lower scores when:
- The text is poorly structured and do not conform to the standard of an English essay or a story.
- The text contains a lot of non-sensical words such as special tokens or programming code.
- The text contains a lot of non-English words.
- The text does not fully answer the writing instruction with full content, and therefore, is unfinished.

Important: Your evaluation output should conform to the following JSON format:
{
  "flow": int,
  "organization": int,
  "balance": int
}

Write <EOE> after outputting the JSON result. | ### Content and Argumentation (30 points)
- **30 points**: The essay presents a clear, well-supported argument that effectively addresses the claim. It includes a variety of strong, relevant evidence and reasoning, and addresses counterarguments comprehensively. (Example: The essay provides historical context, cites specific legal precedents, and includes expert opinions to support its argument.)
- **25 points**: The essay presents a clear argument with strong evidence and reasoning but may lack a few minor details or fail to address a minor counterargument. (Example: The essay provides strong evidence and reasoning but does not address a minor counterargument.)
- **20 points**: …
- **15 points**: …
- **10 points**: The essay presents a weak argument with minimal evidence and reasoning. It may contain significant logical flaws or fail to address the claim effectively. (Example: The essay provides a weak argument and minimal evidence, with significant logical flaws.)
- **0 points**: The essay contains significant factual errors that misinterpret the core business or key aspects of the claim. (Example: The essay incorrectly states that CEMEX is a major sugar producer.)
### Historical and Contextual Analysis (10 points)
…
### Impact Analysis (15 points)
…
### Counterarguments (10 points)
…
### Structure and Coherence (15 points)
…
### Use of Evidence (15 points)
…
### Clarity and Coherence (10 points)
…
### Language and Style (10 points)
…
### Persuasiveness (10 points)
… |

Figure 7: Comparison between hand-crafted and MPO-evolved rubrics for essay writing task.

Expert-Crafted Evaluation Rubric for Essay Writing Task

```
<rubric>
<item>
"Ideas and Content"
Criterion for Integer Score: **6**
Does the writing sample fully accomplish the task (e.g., support an opinion, summarize,
tell a story, or write an article)?  Does it
- present a unifying theme or main idea without going off on tangents?
- stay completely focused on topic and task?
Does the writing sample include thorough, relevant, and complete ideas?  Does it
- include in-depth information and exceptional supporting details that are fully developed?
- fully explore many facets of the topic?

Criterion for Integer Score: **5**
Does the writing sample fully accomplish the task (e.g., support an opinion, summarize,
tell a story, or write an article)?  Does it
- present a unifying theme or main idea without going off on tangents?
- stay focused on topic and task?
Does the writing sample include many relevant ideas?  Does it
- provide in-depth information and more than adequate supporting details that are
developed?
- explore many facets of the topic?

Criterion for Integer Score: **4**
Does the writing sample accomplish the task (e.g., support an opinion, summarize, tell a
story, or write an article)?  Does it
- present a unifying theme or main idea?  (Writing may include minor tangents.)
- stay mostly focused on topic and task?
Does the writing sample include relevant ideas?  Does it
- include sufficient information and supporting details?  (Details may not be fully
developed; ideas may be listed.)
- explore some facets of the topic?
```

```
Criterion for Integer Score: **3**
Does the writing sample minimally accomplish the task (e.g., support an opinion, summarize,
tell a story, or write an article)?  Does it
- attempt a unifying theme or main idea?
- stay somewhat focused on topic and task?
Does the writing sample include some relevant ideas?  Does it
- include some information with only a few details, or list ideas without supporting
details?
- explore some facets of the topic?

Criterion for Integer Score: **2**
Does the writing sample only partially accomplish the task (e.g., support an opinion,
summarize, tell a story, or write an article)?  Does it
- attempt a main idea?
- sometimes lose focus or ineffectively display focus?
Does the writing sample include few relevant ideas?  Does it
- include little information and few or no details?
- explore only one or two facets of the topic?

Criterion for Integer Score: **1**
Does the writing sample fail to accomplish the task (e.g., support an opinion, summarize,
tell a story, or write an article)?  Is it
- difficult for the reader to discern the main idea?
- too brief or too repetitive to establish or maintain a focus?
Does the writing sample include very few relevant ideas?
- Does it include little information with few or no details or unrelated details?
- Is it unsuccessful in attempts to explore any facets of the prompt?
</item>

<item>
"Organization"
Criterion for Integer Score: **6**
Are the ideas in the writing sample organized logically?  Does the writing
- present a meaningful, cohesive whole with a beginning, a middle, and an end (i.e.,
include an inviting introduction and a strong conclusion)?
- progress in an order that enhances meaning?
- include smooth transitions between ideas, sentences, and paragraphs to enhance meaning of
text (i.e., have a clear connection of ideas and use topic sentences)?

Criterion for Integer Score: **5**
Are the ideas in the writing sample organized logically?  Does the writing
- present a meaningful, cohesive whole with a beginning, a middle, and an end (i.e.,
include a solid introduction and conclusion)?
- progress in an order that enhances meaning of text?
- include smooth transitions (e.g., use topic sentences) between sentences and paragraphs
to enhance meaning of text?  (Writing may have an occasional lapse.)

Criterion for Integer Score: **4**
Are the ideas in the writing sample organized logically?  Does the writing
- present a meaningful whole with a beginning, a middle, and an end despite an occasional
lapse (e.g., a weak introduction or conclusion)?
- generally progress in an order that enhances meaning of text?
- include transitions between sentences and paragraphs to enhance meaning of text?
(Transitions may be rough, although some topic sentences are included.)

Criterion for Integer Score: **3**
Is there an attempt to logically organize ideas in the writing sample?  Does the writing
- have a beginning, a middle, or an end that may be weak or absent?
- demonstrate an attempt to progress in an order that enhances meaning?  (Progression of
text may sometimes be unclear or out of order.)
- demonstrate an attempt to include transitions?  (Are some topic sentences used?  Are
transitions between sentences and paragraphs weak or absent?)

Criterion for Integer Score: **2**
Is there a minimal attempt to logically organize ideas in the writing sample?
- Does the writing have only one or two of the three elements:  beginning, middle, and end?
- Is the writing sometimes difficult to follow?  (Progression of text may be confusing or
unclear.)
- Are transitions weak or absent (e.g., few or no topic sentences)?

Criterion for Integer Score: **1**
Are the ideas in the writing sample organized illogically?
- Does it have only one or two of the three elements:  beginning, middle, or end?
- Is it difficult to follow, with the order possibly difficult to discern?
- Are transitions weak or absent (e.g., without topic sentences)?
</item>

<item>
"Style"
```

```
Criterion for Integer Score:  **6**
Does the writing sample exhibit exceptional word usage?  Does it
- include vocabulary to make explanations detailed and precise, descriptions rich, and
actions clear and vivid (e.g., varied word choices, action words, appropriate modifiers,
sensory details)?
- demonstrate control of a challenging vocabulary?
Does the writing sample demonstrate exceptional writing technique?
- Is the writing exceptionally fluent?
- Does it include varied sentence patterns, including complex sentences?
- Does it demonstrate use of writer's techniques (e.g., literary conventions such as
imagery and dialogue and/or literary genres such as humor and suspense)?

Criterion for Integer Score:  **5**
Does the writing sample exhibit very good word usage?  Does it
- include vocabulary to make explanations detailed and precise, descriptions rich, and
actions clear and vivid?
- demonstrate control of vocabulary?
Does the writing sample demonstrate very good writing technique?
- Is the writing very fluent?
- Does it include varied sentence patterns, including complex sentences?
- Does it demonstrate use of writer's techniques (e.g., literary conventions such as
imagery and dialogue and/or literary genres such as humor and suspense)?

Criterion for Integer Score:  **4**
Does the writing sample exhibit good word usage?  Does it
- include vocabulary that is appropriately chosen, with words that clearly convey the
writer's meaning?
- demonstrate control of basic vocabulary?
Does the writing sample demonstrate good writing technique?
- Is the writing fluent?
- Does it exhibit some varied sentence patterns, including some complex sentences?
- Does it demonstrate an attempt to use writer's techniques (e.g., literary conventions
such as imagery and dialogue and/or literary genres such as humor and suspense)?

Criterion for Integer Score:  **3**
Does the writing sample exhibit ordinary word usage?  Does it
- contain basic vocabulary, with words that are predictable and common?
- demonstrate some control of vocabulary?
Does the writing sample demonstrate average writing technique?
- Is the writing generally fluent?
- Does it contain mostly simple sentences (although there may be an attempt at more varied
sentence patterns)?
- Is it generally ordinary and predictable?

Criterion for Integer Score:  **2**
Does the writing sample exhibit minimal word usage?  Does it
- contain limited vocabulary?  (Some words may be used incorrectly.)
- demonstrate minimal control of vocabulary?
Does the writing sample demonstrate minimal writing technique?
- Does the writing exhibit some fluency?
- Does it rely mostly on simple sentences?
- Is it often repetitive, predictable, or dull?

Criterion for Integer Score:  **1**
Does the writing sample exhibit less than minimal word usage?  Does it
- contain limited vocabulary, with many words used incorrectly?
- demonstrate minimal or less than minimal control of vocabulary?
Does the writing sample demonstrate less than minimal writing technique?  Does it
- lack fluency?
- demonstrate problems with sentence patterns?
- consist of writing that is flat and lifeless?
</item>

<item>
"Voice"
Criterion for Integer Score:  **6**
Does the writing sample demonstrate effective adjustment of language and tone to task and
reader?  Does it
- exhibit appropriate register (e.g., formal, personal, or dialect) to suit task?
- demonstrate a strong sense of audience?
- exhibit an original perspective (e.g., authoritative, lively, and/or exciting)?

Criterion for Integer Score:  **5**
Does the writing sample demonstrate effective adjustment of language and tone to task and
reader?  Does it
- exhibit appropriate register (e.g., formal, personal, or dialect) to suit task?
- demonstrate a sense of audience?
- exhibit an original perspective (e.g., authoritative, lively, and/or exciting)?
```

```
Criterion for Integer Score: **4**
Does the writing sample demonstrate an attempt to adjust language and tone to task and
reader?  Does it
- generally exhibit appropriate register (e.g., formal, personal, or dialect) to suit task?
(The writing may occasionally slip out of register.)
- demonstrate some sense of audience?
- attempt an original perspective?

Criterion for Integer Score: **3**
Does the writing sample demonstrate an attempt to adjust language and tone to task and
reader?  Does it
- demonstrate a difficulty in establishing a register (e.g., formal, personal, or dialect)?
- demonstrate little sense of audience?
- generally lack an original perspective?

Criterion for Integer Score: **2**
Does the writing sample demonstrate language and tone that may be inappropriate to task
and reader?  Does it
- demonstrate use of a register inappropriate to the task (e.g., slang or dialect in a
formal setting)?
- demonstrate little or no sense of audience?
- lack an original perspective?

Criterion for Integer Score: **1**
Does the writing sample demonstrate language and tone that may be inappropriate to task
and reader?  Does it
- demonstrate difficulty in choosing an appropriate register?
- demonstrate a lack of a sense of audience?
- lack an original perspective?
</item>

<item>
"Language Conventions"
Criterion for Integer Score: **4**
Does the writing sample exhibit a superior command of language skills?
A Score Point 4 paper exhibits a superior command of written English language conventions.
The paper provides evidence that the student has a thorough control of the concepts
outlined in the Indiana Academic Standards associated with the student's grade level.  In
a Score Point 4 paper, there are no errors that impair the flow of communication.  Errors
are generally of the first-draft variety or occur when the student attempts sophisticated
sentence construction.
- Does the writing sample demonstrate a superior command of capitalization conventions?
- Does the writing sample demonstrate a superior command of the mechanics of punctuation?
- Does the writing sample demonstrate a superior command of grade-level-appropriate
spelling?
- Does the writing sample demonstrate a superior command of grammar and Standard English
usage?
- Does the writing sample demonstrate a superior command of paragraphing?
- Does the writing sample demonstrate a superior command of sentence structure by not using
run-on sentences or sentence fragments?

Criterion for Integer Score: **3**
Score 3:  Does the writing sample exhibit a good control of language skills?
In a Score Point 3 paper, errors are occasional and are often of the first-draft variety;
they have a minor impact on the flow of communication.
- Does the writing sample demonstrate a good control of capitalization conventions?
- Does the writing sample demonstrate a good control of the mechanics of punctuation?
- Does the writing sample demonstrate a good control of grade-level-appropriate spelling?
- Does the writing sample demonstrate a good control of grammar and Standard English usage?
- Does the writing sample demonstrate a good control of paragraphing?
- Does the writing sample demonstrate a good control of sentence structure by only
occasionally using run-on sentences or sentence fragments?

Criterion for Integer Score: **2**
Score 2:  Does the writing sample exhibit a fair control of language skills?
In a Score Point 2 paper, errors are typically frequent and may occasionally impede the
flow of communication.
- Does the writing sample demonstrate a fair control of capitalization conventions?
- Does the writing sample demonstrate a fair control of the mechanics of punctuation?
- Does the writing sample demonstrate a fair control of grade-level-appropriate spelling?
- Does the writing sample demonstrate a fair control of grammar and Standard English usage?
- Does the writing sample demonstrate a fair control of paragraphing?
- Does the writing sample demonstrate a fair control of sentence structure by frequently
using run-on sentences or sentence fragments?

Criterion for Integer Score: **1**
Score 1:  Does the writing sample exhibit a minimal or less than minimal control of
language skills?
In a Score Point 1 paper, errors are serious and numerous.  The reader may need to stop
```

```
and reread part of the sample and may struggle to discern the writer's meaning.
- Does the writing sample demonstrate a minimal control of capitalization conventions?
- Does the writing sample demonstrate a minimal control of the mechanics of punctuation?
- Does the writing sample demonstrate a minimal control of grade-level-appropriate
spelling?
- Does the writing sample demonstrate a minimal control of grammar and Standard English
usage?
- Does the writing sample demonstrate a minimal control of paragraphing?
- Does the writing sample demonstrate a minimal control of sentence structure by using many
run-on sentences or sentence fragments?
</item>

NOTE: The elements of this rubric are applied holistically; no element is intended to
supersede any other element.  The variety and proportion of errors in relation to the
length of the writing sample are considered.  A very brief paper consisting of two or
three sentences may receive no more than 2 score points.
</rubric>
```

### AutoPrompt-Generated Evaluation Rubric for Essay Writing Task

```
<rubric>
<item>
As an expert English teacher, refine your evaluation criteria for scoring student essays
and stories on a scale from 0 to 5.
Assign '0' for essays that are completely incoherent with pervasive errors throughout.
Score '1' for essays with numerous issues but maintain a basic idea or structure.
Give a '2' for essays that have coherence with simplistic content and several errors.
Essays that meet standard narrative and structural expectations but are unremarkable
should receive a '3'.
Award a '4' to essays that exhibit a strong narrative, significant creativity, but may
have a few oversights.
A '5' should be given to essays that not only are free from notable errors but also
possess a distinctive, memorable voice, exceptional creativity, and an engaging narrative
that sets them apart.
Ensure that essays with high-quality narratives, strong imagery, and evocative language
are scored correctly as a '5', to address the previous underestimation in grading.  This
task is a classification class with the following labels:  ["0", "1", "2", "3", "4", "5"].
</item>
</rubric>
```

# I PROMPTS FOR META REWARD MODEL

---

**Meta Analysis**

You are a senior instructor tasked with evaluating a junior instructor's scoring of a
student's generation based on a specific task and prompt instruction.
Your objective is to conduct a meta-level analysis of the junior instructor's evaluation
approach, guiding them in refining their scoring criteria to ensure accurate, nuanced
differentiation between high-quality and subpar generations.
Emphasize strategies for assigning lower scores to undesirable responses and higher scores
to responses that adhere closely to the overall objectives of the task.

The information provided includes:

Task Description:
task_description

Student's Prompt Instructions:
student_prompt

Student's Generation:
student_generation

Junior Instructor's Scoring Criteria:
junior_prompt

Junior Instructor's Assigned Score:
junior_score

Your task:
Critically evaluate the junior instructor's score and justification in relation to the
student's response, task requirements, and prompt instructions.

1. Accuracy of Scoring
 – Determine whether the student's response is receiving an inflated score despite not fully
meeting the task objectives in terms of quality and content.
 – Identify any elements where the response deviates from task expectations, such as
misinterpretation, lack of depth, or overemphasis on irrelevant aspects.

2. Evaluation of Scoring Criteria
 – Assess whether the junior instructor's criteria align with the task's overarching
purpose. Are critical aspects overlooked, or do the criteria require further breakdown
for clarity?
 – Examine whether the distribution of points is logical and correctly sums to the total
score. Flag any inconsistencies and suggest necessary adjustments.

3. Constructive Feedback for Refinement
 – Provide actionable recommendations to enhance the scoring framework, ensuring it is
comprehensive and consistently applied.
 – Emphasize the need for strict penalization in cases of severe errors to maintain
evaluation rigor.

Present the analysis concisely within max_words words. Conclude the response with:
"<EOE>".

Your Analysis:

---

**Meta Refinement**

Based on the meta-level analysis, refine the junior instructor's scoring criteria by
designing an explicit rubric-based framework with separate section items for awarding
points and deducting points.
This rubric must assign specific point values for meeting given criteria, with clear
deductions for any shortcomings.
Fill in any gaps in the existing criteria to cover all relevant aspects of the task.
Provide a concrete example illustrating how the rubric would apply to a typical student
response. Adjust the total score to match the rubric items, ensuring the sum of all
criteria equals the final total.

Use the following structure:

<rubric>
<item>
Score Category Name
 – X1: (Description of the criterion for achieving this score X1, followed by an example.)
 – X2: (Description of the criterion for achieving this score X2, followed by an example.)
 ...
</item>
 ...

---

```
</rubric>

Your generation should be no more than max_words words.  End with "<EOE>".
Important:  You must follow the <rubric> and <item> formatting as shown above.

Junior Instructor's Scoring Criteria (refined):
```

Meta Merging

```
Combine and refine the multiple sets of Junior Instructor's Scoring Criteria into a single,
cohesive set that provides comprehensive guidelines for assessment.

Here are multiple sets of Junior Instructor's Scoring Criteria, delimited by "===":
```
multiple_sets
```

Combine the above concisely without repetition.  The combined criteria should be no more
than max_words words.  Make sure that the points across criteria add up correctly to the
total score.

Use the following structure:

<rubric>
<item>
 Score Category Name
- X1:  (Description of the criterion for achieving this score X1, followed by an example.)
- X2:  (Description of the criterion for achieving this score X2, followed by an example.)
 ...
</item>
 ...
</rubric>

Your generation should be no more than max_words words.  End with "<EOE>".
Important:  You must follow the <rubric> and <item> formatting as shown above.

Junior Instructor's Scoring Criteria (combined):
```

## J    EVOLUTION OF EVALUATION RUBRIC

### J.1    CHANGES IN DISCOURSE MOTIF DISTRIBUTION

In the main body of the paper, we have shown that MPO-evolved rubrics not only lead to higher training rewards but also produce higher-quality generations across the four downstream tasks. To gain deeper insight into the linguistic structure of these rubrics, we apply hierarchical discourse parsing and analyze the resulting discourse subgraphs—referred to as "discourse motifs"—which capture pragmatic discourse relations between textual units ranging from phrases to full paragraphs. This analysis builds on the method introduced by Kim et al. (2024), who used Rhetorical Structure Theory (RST) (Mann & Thompson, 1987) to study discourse patterns in LLM- and human-generated texts. In

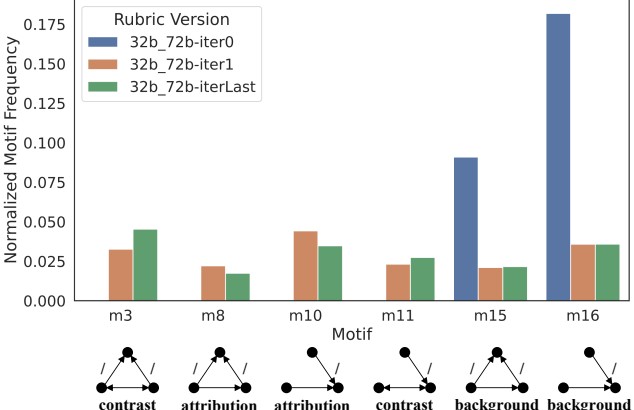

Figure 8: Comparison of discourse motifs found in rubric prompts at the initial stage, after the first, and last MPO step of training for essay writing task.

our setting, we compute the distribution of discourse motifs across three versions of the rubric: the initial version, the version after a single MPO refinement, and the final version at the end of training.

Figure 8 presents bar plots comparing the three rubric versions in terms of discourse motif distributions, with motif types on the x-axis and their normalized frequency within the overall RST graph on the y-axis. The edge label "/" represents a hyperedge relation, typically indicating transitions across

textual levels and signaling that the text exhibits a more hierarchically organized discourse structure. The plot shows that as MPO refinements progress, the evolved rubrics adopt increasingly hierarchical discourse structures, marked by a decrease in Background relations and a corresponding increase in more informative relations such as Contrast and Attribution. Notably, Attribution relations, which explicitly identify the source or ownership of presented information, become more frequent—an important feature for rubric-based evaluation, where attributing claims, reasoning, and judgments clearly is critical for coherent assessment. These trends hint that MPO not only refines content criteria but also implicitly drives the development of richer, more structured evaluation language. We also note that an increase in hierarchical discourse structures is a known characteristic of human-like writing, as reported in Kim et al. (2024; 2025).

## J.2 EXAMPLES

We present a couple of examples of refined evaluation prompts produced during the 32b_32b model training.

### J.2.1 ESSAY WRITING TASK

```
MPO Iteration 1 – Essay Writing

<item>
Task Alignment and Argument Clarity
- 5:  The essay clearly and consistently addresses the specific claim made in the prompt
and presents a well-defined, coherent argument supported by relevant examples and evidence.
(Example:  "The essay argues that the U.S. is not responsible for social backwardness
in Iran because it has not directly influenced Iran's social structures and policies,
supported by specific examples.")
- 4:  The essay mostly addresses the specific claim and presents a coherent argument, but
there are minor inconsistencies or lack of depth.  (Example:  "The essay argues that the
U.S. is not primarily responsible for social backwardness in Iran, but it does not fully
explain how other factors might influence this.")
- 3:  The essay partially addresses the specific claim and presents an argument, but there
are significant inconsistencies or lack of depth.  (Example:  "The essay argues that the
U.S. has not directly caused social backwardness, but it does not fully explain how other
factors might influence this.")
- 2:  The essay addresses the specific claim but the argument is weak and poorly supported.
(Example:  "The essay argues that the U.S. is not responsible, but it lacks supporting
evidence or explanation.")
- 1:  The essay fails to address the specific claim or presents an argument that is
contradictory or irrelevant.  (Example:  "The essay argues that the U.S. has historically
been responsible for promoting social backwardness in Iran.")
- 0:  The essay does not address the specific claim or is completely off-topic.  (Example:
"The essay discusses the history of Iran without mentioning the U.S. or social
backwardness.")
</item>
<item>
Evidence and Reasoning
- 5:  The essay provides strong, relevant evidence and reasoning to support the argument,
with specific examples and data.  (Example:  "The essay cites specific examples of U.S.
policies that did not directly influence Iran's social structures and policies.")
- 4:  The essay provides mostly relevant evidence and reasoning, but some points are weak
or not fully supported.  (Example:  "The essay cites examples of U.S. policies but does
not fully explain how they do not contribute to social backwardness.")
- 3:  The essay provides some relevant evidence and reasoning, but there are significant
gaps or weak points.  (Example:  "The essay mentions U.S. policies but does not provide
specific examples or explanations.")
- 2:  The essay provides weak or irrelevant evidence and reasoning.  (Example:  "The
essay mentions U.S. policies but does not explain how they are unrelated to social
backwardness.")
- 1:  The essay does not provide any evidence or reasoning to support the argument.
(Example:  "The essay makes claims without providing any supporting evidence.")
- 0:  The essay provides evidence and reasoning that contradicts the argument.  (Example:
"The essay provides examples of U.S. policies that contributed to social backwardness.")
</item>
<item>
Counterarguments and Refutation
- 5:  The essay addresses potential counterarguments and provides strong refutations.
(Example:  "The essay acknowledges that U.S. policies might have some indirect influence
but explains why this does not make the U.S. responsible for social backwardness.")
- 4:  The essay addresses some counterarguments and provides mostly strong refutations.
(Example:  "The essay acknowledges some counterarguments but does not fully refute them.")
- 3:  The essay addresses some counterarguments but provides weak or incomplete
refutations.  (Example:  "The essay mentions counterarguments but does not fully address
```

```
them.")
 - 2:  The essay addresses some counterarguments but does not provide any refutations.
 (Example:  "The essay mentions counterarguments without explaining why they do not
 undermine the argument.")
 - 1:  The essay does not address any counterarguments.  (Example:  "The essay does not
 mention any counterarguments.")
 - 0:  The essay addresses counterarguments but fails to refute them or provides weak
 refutations.  (Example:  "The essay mentions counterarguments but does not provide any
 refutations or provides weak refutations.")
 </item>
 <item>
 Structure and Coherence
 - 5:  The essay is well-organized and coherent, with a clear structure and logical flow.
 (Example:  "The essay has an introduction, body paragraphs, and a conclusion that are
 logically connected.")
 - 4:  The essay has a mostly clear structure, but some points may be slightly disconnected
 or lack smooth transitions.  (Example:  "The essay has a clear structure but some points
 are not well-connected or lack transitions.")
 - 3:  The essay has a somewhat clear structure, but the points are not well-connected,
 and the transitions are weak.  (Example:  "The essay has a structure but the points are
 disjointed and the transitions are weak.")
 - 2:  The essay lacks a clear structure, and the points are disjointed and disconnected.
 (Example:  "The essay has no clear structure and the points are not well-connected.")
 - 1:  The essay is poorly structured, with no clear introduction, body, or conclusion.
 (Example:  "The essay lacks a clear structure and the points are entirely disjointed.")
 - 0:  The essay is completely disorganized and lacks any structure.  (Example:  "The essay
 is a collection of unrelated points with no structure.")
 </item> <item>
 Depth of Analysis
 - 5:  The essay demonstrates a deep and nuanced analysis of the issue, addressing
 complexities and providing specific examples and data.  (Example:  "The essay discusses
 the historical roots of the conflict, the cultural identity of Palestinians, and the
 potential repercussions of the proposed solution.")
 - 4:  The essay provides some depth and nuance in the analysis but lacks specific examples
 or data.  (Example:  "The essay mentions that the conflict is about identity and survival
 but does not provide substantial evidence or reasoning.")
 - 3:  The essay provides minimal depth and nuance in the analysis, lacking specific
 examples or data.  (Example:  "The essay mentions the conflict is about identity but
 does not provide substantial historical or cultural context.")
 - 2:  The essay lacks depth and nuance in the analysis, providing vague statements without
 specific examples or data.  (Example:  "The essay is vague and does not provide any
 substantial argument or reasoning.")
 - 1:  The essay lacks any meaningful analysis or is superficial.  (Example:  "The essay
 does not provide any meaningful analysis and remains superficial in its discussion.")
 - 0:  The essay is completely lacking in analysis and is purely superficial.  (Example:
 "The essay is vague and does not provide any substantial content or context.")
 </item>
 <item>
 Language and Mechanics
 - 5:  The essay demonstrates clear and effective use of language with minimal grammatical
 errors.  (Example:  "The essay uses clear and precise language with no significant
 grammatical errors.")
 - 4:  The essay has some grammatical errors but is still generally clear.  (Example:  "The
 essay has a few grammatical errors but is still understandable.")
 - 3:  The essay has some grammatical errors that affect clarity but is still mostly
 comprehensible.  (Example:  "The essay has some errors in grammar or style but is still
 generally clear.")
 - 2:  The essay has significant grammatical errors that affect clarity and coherence.
 (Example:  "The essay has multiple grammatical or stylistic errors that make it difficult
 to understand.")
 - 1:  The essay has significant grammatical errors that severely affect clarity and
 coherence.  (Example:  "The essay has multiple grammatical or stylistic errors that make
 it difficult to understand.")
 - 0:  The essay is completely unclear and incoherent due to significant grammatical or
 stylistic errors.  (Example:  "The essay is difficult to read due to numerous grammatical
 or stylistic errors.")
 </item>
```

## MPO Iteration 40 – Essay Writing

```
<item>
Task Alignment and Argument Clarity (10 points)
 - 10:  The essay clearly and consistently addresses the specific claim made in the prompt
 and presents a well-defined, coherent argument supported by relevant examples and evidence.
 The argument is fully developed and addresses the core issue of the prompt.  (Example:
 "The essay argues that higher education is counterproductive by providing specific
 examples of student debt, income gaps, and over-specialization in certain fields.")
 - 8:  The essay mostly addresses the specific claim and presents a coherent argument, but
 there are minor inconsistencies or lack of depth.  The argument is somewhat developed but
```

```
could benefit from more specific examples or a deeper analysis. (Example: "The essay
argues that higher education is counterproductive but lacks substantial examples or a
clear connection to the context.")
 - 6:  The essay partially addresses the specific claim and presents an argument, but
there are significant inconsistencies or lack of depth. The argument is weak and lacks
substantial examples or a clear connection to the core issue. (Example: "The essay
argues that higher education is counterproductive but fails to address the specific
aspects of student debt, income gaps, and over-specialization.")
 - 4:  The essay addresses the specific claim but the argument is weak and poorly supported.
The argument lacks substantial examples or a clear connection to the core issue. (Example:
"The essay argues that higher education is counterproductive but lacks supporting evidence
or explanation.")
 - 2:  The essay fails to address the specific claim or presents an argument that is
contradictory or irrelevant. The argument is not aligned with the prompt and does not
address the core issue. (Example: "The essay argues that higher education is beneficial,
which contradicts the prompt.")
 - 0:  The essay does not address the specific claim or is completely off-topic. The
argument is not related to the prompt and does not address the core issue. (Example:
"The essay discusses unrelated topics without addressing the claim.")
</item>
<item>
Evidence and Reasoning (10 points)
 - 10:  The essay provides strong, relevant evidence and reasoning to support the argument,
with specific examples and data from current events and historical contexts. The evidence
is well-connected to the core issue and supports the argument effectively. (Example:
"The essay cites specific studies showing the burden of student debt, the widening
income gap, and the over-specialization in certain fields, providing detailed examples
and data.")
 - 8:  The essay provides mostly relevant evidence and reasoning, but some points are
weak or not fully supported. The evidence is somewhat connected to the core issue but
could benefit from more substantial examples or a deeper analysis. (Example: "The essay
mentions studies but does not provide substantial evidence or explanation.")
 - 6:  The essay provides some relevant evidence and reasoning, but there are significant
gaps or weak points. The evidence is not well-connected to the core issue and lacks
substantial examples or a clear connection to the context. (Example: "The essay mentions
studies but does not provide specific examples or explanations.")
 - 4:  The essay provides weak or irrelevant evidence and reasoning. The evidence is not
well-connected to the core issue and lacks substantial examples or a clear connection
to the context. (Example: "The essay mentions studies but does not explain how this
supports the argument or provide substantial evidence.")
 - 2:  The essay does not provide any evidence or reasoning to support the argument. The
evidence is not related to the core issue and lacks substantial examples or a clear
connection to the context. (Example: "The essay makes claims without providing any
supporting evidence.")
 - 0:  The essay provides evidence and reasoning that contradict the argument. The evidence
is not related to the core issue and lacks substantial examples or a clear connection to
the context. (Example: "The essay provides examples that support the claim that higher
education is beneficial.")
</item>
<item>
Counterarguments and Refutation (10 points)
 - 10:  The essay addresses potential counterarguments and provides strong refutations.
The counterarguments are acknowledged and effectively refuted with specific examples
and reasoning. (Example: "The essay acknowledges that some argue higher education is
beneficial by explaining the burden of student debt, the widening income gap, and the
over-specialization in certain fields.")
 - 8:  The essay addresses some counterarguments and provides mostly strong refutations.
The counterarguments are acknowledged but not fully refuted. (Example: "The essay
acknowledges some counterarguments but does not fully refute them.")
 - 6:  The essay addresses some counterarguments but provides weak or incomplete
refutations. The counterarguments are acknowledged but not fully addressed. (Example:
"The essay mentions counterarguments but does not fully address them.")
 - 4:  The essay addresses some counterarguments but does not provide any refutations. The
counterarguments are acknowledged but not addressed. (Example: "The essay mentions
counterarguments without explaining why they do not undermine the argument.")
 - 2:  The essay does not address any counterarguments. The counterarguments are not
acknowledged or addressed. (Example: "The essay does not mention any counterarguments.")
 - 0:  The essay addresses counterarguments but fails to refute them or provides weak
refutations. The counterarguments are acknowledged but not effectively refuted. (Example:
"The essay mentions counterarguments but does not provide any refutations or provides weak
refutations.")
</item>
<item>
Structure and Coherence (10 points)
 - 10:  The essay is well-organized and coherent, with a clear structure and logical flow.
The introduction sets up the argument, body paragraphs provide evidence and reasoning,
and the conclusion summarizes the argument and provides a final thought. (Example:
"The essay has an introduction that sets up the argument, body paragraphs that provide
evidence and reasoning, and a conclusion that summarizes the argument and provides a final
```

```
 thought.")
 - 8:  The essay has a mostly clear structure, but some points may be slightly disconnected
 or lack smooth transitions.  The introduction sets up the argument, body paragraphs
 provide evidence and reasoning, and the conclusion summarizes the argument but lacks
 smooth transitions.  (Example:  "The essay has a clear structure but some points are not
 well-connected or lack transitions.")
 - 6:  The essay has a somewhat clear structure, but the points are not well-connected,
 and the transitions are weak.  The introduction sets up the argument, body paragraphs
 provide evidence and reasoning, but the transitions are weak.  (Example:  "The essay has a
 structure but the points are disjointed and the transitions are weak.")
 - 4:  The essay lacks a clear structure, and the points are disjointed and disconnected.
 The introduction sets up the argument, but the body paragraphs are disjointed and the
 conclusion is unclear.  (Example:  "The essay has no clear structure and the points are
 not well-connected.")
 - 2:  The essay is poorly structured, with no clear introduction, body, or conclusion.  The
 essay lacks a clear structure and the points are entirely disjointed.  (Example:  "The
 essay lacks a clear structure and the points are entirely disjointed.")
 - 0:  The essay is completely disorganized and lacks any structure.  The essay is a
 collection of unrelated points with no structure.  (Example:  "The essay is a collection
 of unrelated points with no structure.")
 </item>
 <item>
 Depth of Analysis (10 points)
 - 10:  The essay demonstrates a deep and nuanced analysis of the issue, addressing
 complexities and providing specific examples and data from current events and historical
 contexts.  The analysis is well-connected to the core issue and supports the argument
 effectively.  (Example:  "The essay discusses the complexities of higher education,
 providing specific examples of student debt, income gaps, and over-specialization in
 certain fields, and provides detailed data from reliable sources.")
 - 8:  The essay provides some depth and nuance in the analysis but lacks specific examples
 or data.  The analysis is somewhat connected to the core issue but could benefit from more
 substantial examples or a deeper analysis.  (Example:  "The essay mentions that higher
 education has issues but does not provide substantial evidence or reasoning.")
 - 6:  The essay provides minimal depth and nuance in the analysis, lacking specific
 examples or data.  The analysis is not well-connected to the core issue and lacks
 substantial examples or a clear connection to the context.  (Example:  "The essay mentions
 that higher education has issues but does not provide specific examples or explanations.")
 - 4:  The essay lacks depth and nuance in the analysis, providing vague statements without
 specific examples or data.  The analysis is not well-connected to the core issue and lacks
 substantial examples or a clear connection to the context.  (Example:  "The essay is vague
 and does not provide any substantial argument or reasoning.")
 - 2:  The essay lacks any meaningful analysis or is superficial.  The analysis is not
 related to the core issue and lacks substantial examples or a clear connection to the
 context.  (Example:  "The essay does not provide any meaningful analysis and remains
 superficial in its discussion.")
 - 0:  The essay is completely lacking in analysis and is purely superficial.  The analysis
 is not related to the core issue and lacks substantial examples or a clear connection to
 the context.  (Example:  "The essay is vague and does not provide any substantial content
 or context.")
 </item>
 <item>
 Language and Mechanics (10 points)
 - 10:  The essay demonstrates clear and effective use of language with minimal grammatical
 errors.  The writing is clear, coherent, and free of significant errors that affect
 clarity or coherence.  (Example:  "The essay uses clear and precise language with no
 significant grammatical errors.")
 - 8:  The essay has some grammatical errors but is still generally clear.  The writing
 is mostly clear and coherent, with a few minor errors that do not significantly
 affect clarity.  (Example:  "The essay has a few grammatical errors but is still
 understandable.")
 - 6:  The essay has some grammatical errors that affect clarity but is still mostly
 comprehensible.  The writing is generally clear but has some errors that affect clarity.
 (Example:  "The essay has some errors in grammar or style but is still generally clear.")
 - 4:  The essay has significant grammatical errors that affect clarity and coherence.  The
 writing is unclear and difficult to follow due to significant errors.  (Example:  "The
 essay has multiple grammatical or stylistic errors that make it difficult to understand.")
 - 2:  The essay has significant grammatical errors that severely affect clarity and
 coherence.  The writing is unclear and difficult to follow due to multiple significant
 errors.  (Example:  "The essay has multiple grammatical or stylistic errors that make it
 difficult to understand.")
 - 0:  The essay is completely unclear and incoherent due to significant grammatical
 or stylistic errors.  The writing is difficult to read due to numerous grammatical or
 stylistic errors.  (Example:  "The essay is difficult to read due to numerous grammatical
 or stylistic errors.")
 </item>
```

### J.2.2 ETHICAL REASONING

---

**MPO Iteration 1 – Ethical Reasoning**

```
<item>
Logical Coherence (0-5)
- 5: The reasoning is logically consistent, free from contradictions, and well-supported
by evidence or examples.  The argument is clear, structured, and avoids repetition or
disjointedness.
- Example:  "The girlfriend's reaction is wrong because she overreacted.  This is not
a justifiable response as it lacks empathy and understanding.  The act of forgetting
to defrost the shrimp is a minor inconvenience, and the girlfriend's response is
disproportionate, disrespectful, and potentially harmful to the relationship.  Trusting
someone with your body is a significant decision, and I believe it's important to
prioritize safety and quality.  <verdict>WRONG</verdict>"
- 4: The reasoning is mostly logical, with a few minor contradictions or gaps in
reasoning.  The argument is coherent but could be more clear or structured.
- Example:  "The girlfriend's reaction is wrong because she overreacted.  This is not
a justifiable response as it lacks empathy and understanding.  The act of forgetting
to defrost the shrimp is a minor inconvenience, and the girlfriend's response is
disproportionate and disrespectful.  <verdict>WRONG</verdict>"
- 3: The reasoning is somewhat logical but contains some contradictions or gaps in
reasoning.  The argument is coherent but could be more clear or structured.
- Example:  "The girlfriend's reaction is wrong because she overreacted.  This is not
a justifiable response as it lacks empathy and understanding.  The act of forgetting
to defrost the shrimp is a minor inconvenience, and the girlfriend's response is
disproportionate.  <verdict>WRONG</verdict>"
- 2: The reasoning is partially logical, with significant contradictions or gaps in
reasoning.  The argument is somewhat coherent but lacks clarity and structure.
- Example:  "The girlfriend's reaction is wrong because she overreacted.  This is not a
justifiable response as it lacks empathy and understanding.  The act of forgetting to
defrost the shrimp is a minor inconvenience.  <verdict>WRONG</verdict>"
- 1: The reasoning is illogical, with major contradictions or gaps in reasoning.  The
argument is incoherent and lacks clarity.
- Example:  "The girlfriend's reaction is wrong because she overreacted.  The act of
forgetting to defrost the shrimp is a minor inconvenience.  <verdict>WRONG</verdict>"
- 0: The reasoning is completely illogical, with no coherent argument or supporting
evidence.  The argument is incoherent and lacks clarity.
- Example:  "The girlfriend's reaction is wrong.  <verdict>WRONG</verdict>"
</item>
<item>
Depth of Moral Insight (0-5)
- 5:  The analysis demonstrates a deep understanding of the ethical issues involved,
providing nuanced insights and thoughtful consideration of multiple perspectives.  The
reasoning explores multiple ethical principles and their implications.
- Example:  "The girlfriend's reaction is wrong because she overreacted.  This is not
a justifiable response as it lacks empathy and understanding.  The act of forgetting
to defrost the shrimp is a minor inconvenience, and the girlfriend's response is
disproportionate, disrespectful, and potentially harmful to the relationship.  This
response reflects a lack of consideration for others and a disregard for the impact
of one's actions.  It also violates deontological principles, which emphasize the
importance of treating others with respect and dignity.  Additionally, her response fails
to consider the utilitarian principle of maximizing overall happiness and well-being.
<verdict>WRONG</verdict>"
- 4:  The analysis demonstrates a good understanding of the ethical issues involved, with
some thoughtful consideration of multiple perspectives.  The reasoning explores some
ethical principles and their implications.
- Example:  "The girlfriend's reaction is wrong because she overreacted.  This is not
a justifiable response as it lacks empathy and understanding.  The act of forgetting
to defrost the shrimp is a minor inconvenience, and the girlfriend's response is
disproportionate, disrespectful, and potentially harmful to the relationship.  This
response reflects a lack of consideration for others and a disregard for the impact of
one's actions.  It also violates deontological principles, which emphasize the importance
of treating others with respect and dignity.  <verdict>WRONG</verdict>"
- 3:  The analysis demonstrates a basic understanding of the ethical issues involved,
with some consideration of multiple perspectives.  The reasoning touches on some ethical
principles and their implications.
- Example:  "The girlfriend's reaction is wrong because she overreacted.  This is not
a justifiable response as it lacks empathy and understanding.  The act of forgetting
to defrost the shrimp is a minor inconvenience, and the girlfriend's response is
disproportionate and disrespectful.  <verdict>WRONG</verdict>"
- 2:  The analysis demonstrates a limited understanding of the ethical issues involved,
with little consideration of multiple perspectives.  The reasoning touches on some ethical
principles but lacks depth.
- Example:  "The girlfriend's reaction is wrong because she overreacted.  This is not a
justifiable response as it lacks empathy and understanding.  The act of forgetting to
defrost the shrimp is a minor inconvenience.  <verdict>WRONG</verdict>"
- 1:  The analysis demonstrates a minimal understanding of the ethical issues involved,
with no consideration of multiple perspectives.  The reasoning touches on some ethical
```

```
principles but lacks depth.
- Example:  "The girlfriend's reaction is wrong because she overreacted.  This is not a
justifiable response as it lacks empathy and understanding.  <verdict>WRONG</verdict>"
- 0:  The analysis demonstrates no understanding of the ethical issues involved.  The
reasoning lacks any exploration of ethical principles or ethical theories.
- Example:  "The girlfriend's reaction is wrong because she overreacted.
<verdict>WRONG</verdict>"
</item>
<item>
Alignment with Established Ethical Principles (0-5)
- 5:  The response references and applies relevant ethical principles or theories
comprehensively and accurately.  The reasoning explicitly identifies and applies specific
ethical theories or principles, including how they apply to the specific situation and why
they are relevant.
- Example:  "The girlfriend's reaction is wrong because she overreacted.  This is not
a justifiable response as it lacks empathy and understanding.  The act of forgetting
to defrost the shrimp is a minor inconvenience, and the girlfriend's response is
disproportionate, disrespectful, and potentially harmful to the relationship.  This
response reflects a lack of consideration for others and a disregard for the impact
of one's actions.  It also violates deontological principles, which emphasize the
importance of treating others with respect and dignity.  Additionally, her response fails
to consider the utilitarian principle of maximizing overall happiness and well-being.
<verdict>WRONG</verdict>"
- 4:  The response references and applies relevant ethical principles or theories with
some accuracy.  The reasoning references some ethical principles or theories but does not
provide a clear explanation of how they apply to the situation.
- Example:  "The girlfriend's reaction is wrong because she overreacted.  This is not
a justifiable response as it lacks empathy and understanding.  The act of forgetting
to defrost the shrimp is a minor inconvenience, and the girlfriend's response is
disproportionate, disrespectful, and potentially harmful to the relationship.  This
response reflects a lack of consideration for others and a disregard for the impact of
one's actions.  It also violates deontological principles, which emphasize the importance
of treating others with respect and dignity.  <verdict>WRONG</verdict>"
- 3:  The response references some ethical principles or theories but with limited
accuracy.  The reasoning references some ethical principles or theories but does not
provide a clear explanation of how they apply to the situation.
- Example:  "The girlfriend's reaction is wrong because she overreacted.  This is not
a justifiable response as it lacks empathy and understanding.  The act of forgetting
to defrost the shrimp is a minor inconvenience, and the girlfriend's response is
disproportionate and disrespectful.  <verdict>WRONG</verdict>"
- 2:  The response references some ethical principles or theories but with significant
inaccuracies.  The reasoning references some ethical principles or theories but does not
provide a clear explanation of how they apply to the situation.
- Example:  "The girlfriend's reaction is wrong because she overreacted.  This is not a
justifiable response as it lacks empathy and understanding.  The act of forgetting to
defrost the shrimp is a minor inconvenience.  <verdict>WRONG</verdict>"
- 1:  The response references relevant ethical principles or theories but with major
inaccuracies.  The reasoning references some ethical principles or theories but does not
provide a clear explanation of how they apply to the situation.
- Example:  "The girlfriend's reaction is wrong because she overreacted.  This is not a
justifiable response as it lacks empathy and understanding.  <verdict>WRONG</verdict>"
- 0:  The response does not reference any relevant ethical principles or theories.  The
reasoning does not reference any ethical principles or theories.
- Example:  "The girlfriend's reaction is wrong.  <verdict>WRONG</verdict>"
</item>
<item>
Clarity of Verdict (0-5)
- 5:  The final verdict is clear, properly formatted, and well-supported by the ethical
reasoning.  The verdict is clearly stated within the <verdict></verdict> tags, and the
reasoning supports the verdict.
- Example:  "The girlfriend's reaction is wrong because she overreacted.  This is not
a justifiable response as it lacks empathy and understanding.  The act of forgetting
to defrost the shrimp is a minor inconvenience, and the girlfriend's response is
disproportionate, disrespectful, and potentially harmful to the relationship.  This
response reflects a lack of consideration for others and a disregard for the impact
of one's actions.  It also violates deontological principles, which emphasize the
importance of treating others with respect and dignity.  Additionally, her response fails
to consider the utilitarian principle of maximizing overall happiness and well-being.
<verdict>WRONG</verdict>"
- 4:  The final verdict is clear and properly formatted but lacks some supporting evidence.
The verdict is clearly stated within the <verdict></verdict> tags, but the reasoning does
not fully support the verdict.
- Example:  "The girlfriend's reaction is wrong because she overreacted.  This is not
a justifiable response as it lacks empathy and understanding.  The act of forgetting
to defrost the shrimp is a minor inconvenience, and the girlfriend's response is
disproportionate, disrespectful, and potentially harmful to the relationship.  This
response reflects a lack of consideration for others and a disregard for the impact of
one's actions.  It also violates deontological principles, which emphasize the importance
of treating others with respect and dignity.  <verdict>WRONG</verdict>"
```

```
- 3:  The final verdict is somewhat clear and properly formatted but lacks supporting
evidence.  The verdict is clearly stated within the <verdict></verdict> tags, but the
reasoning does not fully support the verdict.
- Example:  "The girlfriend's reaction is wrong because she overreacted.  This is not
a justifiable response as it lacks empathy and understanding.  The act of forgetting
to defrost the shrimp is a minor inconvenience, and the girlfriend's response is
disproportionate and disrespectful.  <verdict>WRONG</verdict>"
- 2:  The final verdict is partially clear and properly formatted but lacks significant
supporting evidence.  The verdict is clearly stated within the <verdict></verdict> tags,
but the reasoning does not fully support the verdict.
- Example:  "The girlfriend's reaction is wrong because she overreacted.  This is not a
justifiable response as it lacks empathy and understanding.  The act of forgetting to
defrost the shrimp is a minor inconvenience.  <verdict>WRONG</verdict>"
- 1:  The final verdict is unclear or improperly formatted.  The verdict is not clearly
stated within the <verdict></verdict> tags, and the reasoning does not support the
verdict.
- Example:  "The girlfriend's reaction is wrong because she overreacted.  This is not a
justifiable response as it lacks empathy and understanding.  <verdict>WRONG</verdict>"
- 0:  The final verdict is completely unclear or improperly formatted.  The verdict is not
clearly stated within the <verdict></verdict> tags, and the reasoning does not support the
verdict.
- Example:  "The girlfriend's reaction is wrong.  <verdict>WRONG</verdict>"
</item>
```

### MPO Iteration 20 – Ethical Reasoning

```
<item>
Logical Coherence (0-5)
- 5:  The reasoning is logically consistent, free from contradictions, and well-supported
by evidence or examples.  The argument is clear, structured, and avoids repetition
or disjointedness.  It considers multiple perspectives and the broader ethical
implications of the action.  The verdict is presented within the required tags.
<verdict>RIGHT</verdict>
- Example:  "The action of not using the GPS app is justified.  It respects personal
autonomy and privacy, which are fundamental human rights.  The decision to not use the app
is a reasonable response to concerns about personal privacy and autonomy.  The reasoning
is clear, structured, and avoids repetition.  <verdict>RIGHT</verdict>"
- 4:  The reasoning is mostly logical, with a few minor contradictions or gaps in
reasoning.  The argument is coherent but could be more clear or structured.  The reasoning
considers some perspectives but lacks a comprehensive analysis.  The verdict is presented
within the required tags.  <verdict>RIGHT</verdict>
- Example:  "The action of not using the GPS app is justified.  It respects personal
autonomy and privacy, which are fundamental human rights.  The decision to not use the app
is a reasonable response to concerns about personal privacy and autonomy.  The reasoning
is mostly coherent but could be more clear or structured.  <verdict>RIGHT</verdict>"
- 3:  The reasoning is somewhat logical but contains some contradictions or gaps in
reasoning.  The argument is coherent but could be more clear or structured.  The reasoning
touches on some perspectives but lacks a comprehensive analysis.  The verdict is presented
within the required tags.  <verdict>RIGHT</verdict>
- Example:  "The action of not using the GPS app is justified.  It respects personal
autonomy and privacy, which are fundamental human rights.  The decision to not use the app
is a reasonable response to concerns about personal privacy and autonomy.  The reasoning
is somewhat coherent but lacks clarity.  <verdict>RIGHT</verdict>"
- 2:  The reasoning is partially logical, with significant contradictions or gaps in
reasoning.  The argument is somewhat coherent but lacks clarity and structure.  The
reasoning is disjointed and lacks coherence.  The verdict is presented within the required
tags.  <verdict>RIGHT</verdict>
- Example:  "The action of not using the GPS app is justified.  It respects personal
autonomy and privacy, which are fundamental human rights.  The decision to not use the app
is a reasonable response to concerns about personal privacy and autonomy.  The reasoning
is disjointed and lacks coherence.  <verdict>RIGHT</verdict>"
- 1:  The reasoning is illogical, with major contradictions or gaps in reasoning.  The
argument is incoherent and lacks clarity.  The reasoning is incoherent and lacks logical
structure.  The verdict is presented within the required tags.  <verdict>RIGHT</verdict>
- Example:  "The action of not using the GPS app is justified.  It respects personal
autonomy and privacy, which are fundamental human rights.  The decision to not use the app
is a reasonable response to concerns about personal privacy and autonomy.  The reasoning
is incoherent and lacks logical structure.  <verdict>RIGHT</verdict>"
- 0:  The reasoning is completely illogical, with no coherent argument or supporting
evidence.  The argument is incoherent and lacks clarity.  The reasoning is completely
illogical, with no coherent argument or supporting evidence.  The verdict is presented
within the required tags.  <verdict>RIGHT</verdict>
- Example:  "The action of not using the GPS app is justified.  <verdict>RIGHT</verdict>"
</item>
<item>
Depth of Moral Insight (0-5)
- 5:  The analysis demonstrates a deep understanding of the ethical issues involved,
providing nuanced insights and thoughtful consideration of multiple perspectives.  The
reasoning explores multiple ethical principles and their implications, including the
```

```
balance between personal needs and others' well-being.  The reasoning explores the ethical
principles of personal safety, respect for autonomy, and the impact on the relationship.
The verdict is presented within the required tags.  <verdict>RIGHT</verdict>
 - Example: "The action of not using the GPS app is justified.  It respects personal
autonomy and privacy, which are fundamental human rights.  The decision to not use the app
is a reasonable response to concerns about personal privacy and autonomy.  The reasoning
explores the ethical principles of personal safety, respect for autonomy, and the impact
on the relationship.  <verdict>RIGHT</verdict>"
 - 4:  The analysis demonstrates a good understanding of the ethical issues involved, with
some thoughtful consideration of multiple perspectives.  The reasoning explores some
ethical principles and their implications.  The reasoning considers the ethical principles
of personal safety and respect for autonomy.  The verdict is presented within the required
tags.  <verdict>RIGHT</verdict>
 - Example: "The action of not using the GPS app is justified.  It respects personal
autonomy and privacy, which are fundamental human rights.  The decision to not use
the app is a reasonable response to concerns about personal privacy and autonomy.  The
reasoning considers the ethical principles of personal safety and respect for autonomy.
<verdict>RIGHT</verdict>"
 - 3:  The analysis demonstrates a basic understanding of the ethical issues involved,
with some consideration of multiple perspectives.  The reasoning touches on some ethical
principles and their implications.  The reasoning touches on the ethical principles of
personal safety and respect for autonomy.  The verdict is presented within the required
tags.  <verdict>RIGHT</verdict>
 - Example: "The action of not using the GPS app is justified.  It respects personal
autonomy and privacy, which are fundamental human rights.  The decision to not use
the app is a reasonable response to concerns about personal privacy and autonomy.  The
reasoning touches on the ethical principles of personal safety and respect for autonomy.
<verdict>RIGHT</verdict>"
 - 2:  The analysis demonstrates a limited understanding of the ethical issues involved,
with little consideration of multiple perspectives.  The reasoning touches on some ethical
principles but lacks depth.  The reasoning touches on the ethical principles of personal
safety and respect for autonomy but lacks depth.  The verdict is presented within the
required tags.  <verdict>RIGHT</verdict>
 - Example: "The action of not using the GPS app is justified.  It respects personal
autonomy and privacy, which are fundamental human rights.  The decision to not use the app
is a reasonable response to concerns about personal privacy and autonomy.  The reasoning
touches on the ethical principles of personal safety and respect for autonomy but lacks
depth.  <verdict>RIGHT</verdict>"
 - 1:  The analysis demonstrates a minimal understanding of the ethical issues involved,
with no consideration of multiple perspectives.  The reasoning touches on some ethical
principles but lacks depth.  The reasoning touches on the ethical principles of personal
safety and respect for autonomy but lacks depth.  The verdict is presented within the
required tags.  <verdict>RIGHT</verdict>
 - Example: "The action of not using the GPS app is justified.  It respects personal
autonomy and privacy, which are fundamental human rights.  The decision to not use the app
is a reasonable response to concerns about personal privacy and autonomy.  The reasoning
touches on the ethical principles of personal safety and respect for autonomy but lacks
depth.  <verdict>RIGHT</verdict>"
 - 0:  The analysis demonstrates no understanding of the ethical issues involved.  The
reasoning lacks any exploration of ethical principles or ethical theories.  The reasoning
lacks any exploration of ethical principles or ethical theories.  The verdict is presented
within the required tags.  <verdict>RIGHT</verdict>
 - Example:  "The action of not using the GPS app is justified.  <verdict>RIGHT</verdict>"
</item>
<item>
Alignment with Established Ethical Principles (0-5)
 - 5:  The response references and applies relevant ethical principles or theories
comprehensively and accurately.  The reasoning explicitly identifies and applies
specific ethical principles or theories, including how they apply to the specific
situation and why they are relevant.  The reasoning explicitly identifies and applies
specific ethical principles such as personal safety, respect for autonomy, and the
impact on the relationship.  The verdict is presented within the required tags.
<verdict>RIGHT</verdict>
 - Example: "The action of not using the GPS app is justified.  It respects personal
autonomy and privacy, which are fundamental human rights.  The decision to not use the app
is a reasonable response to concerns about personal privacy and autonomy.  The reasoning
explicitly identifies and applies specific ethical principles such as personal safety,
respect for autonomy, and the impact on the relationship.  <verdict>RIGHT</verdict>"
 - 4:  The response references and applies relevant ethical principles or theories with
some accuracy.  The reasoning references some ethical principles or theories but does not
provide a clear explanation of how they apply to the situation.  The reasoning references
some ethical principles such as personal safety and respect for autonomy but does not
provide a clear explanation of how they apply to the situation.  The verdict is presented
within the required tags.  <verdict>RIGHT</verdict>
 - Example: "The action of not using the GPS app is justified.  It respects personal
autonomy and privacy, which are fundamental human rights.  The decision to not use
the app is a reasonable response to concerns about personal privacy and autonomy.  The
reasoning references some ethical principles such as personal safety and respect for
autonomy but does not provide a clear explanation of how they apply to the situation.
<verdict>RIGHT</verdict>"
```

```
- 3:  The response references some ethical principles or theories but with limited
 accuracy.  The reasoning references some ethical principles or theories but does not
 provide a clear explanation of how they apply to the situation.  The reasoning references
 some ethical principles such as personal safety and respect for autonomy but does not
 provide a clear explanation of how they apply to the situation.  The verdict is presented
 within the required tags.  <verdict>RIGHT</verdict>
- Example:  "The action of not using the GPS app is justified.  It respects personal
 autonomy and privacy, which are fundamental human rights.  The decision to not use
 the app is a reasonable response to concerns about personal privacy and autonomy.  The
 reasoning references some ethical principles such as personal safety and respect for
 autonomy but does not provide a clear explanation of how they apply to the situation.
 <verdict>RIGHT</verdict>"
- 2:  The response references some ethical principles or theories but with significant
 inaccuracies.  The reasoning references some ethical principles or theories but does not
 provide a clear explanation of how they apply to the situation.  The reasoning references
 some ethical principles such as personal safety and respect for autonomy but does not
 provide a clear explanation of how they apply to the situation.  The verdict is presented
 within the required tags.  <verdict>RIGHT</verdict>
- Example:  "The action of not using the GPS app is justified.  It respects personal
 autonomy and privacy, which are fundamental human rights.  The decision to not use
 the app is a reasonable response to concerns about personal privacy and autonomy.  The
 reasoning references some ethical principles such as personal safety and respect for
 autonomy but does not provide a clear explanation of how they apply to the situation.
 <verdict>RIGHT</verdict>"
- 1:  The response references relevant ethical principles or theories but with major
 inaccuracies.  The reasoning references some ethical principles or theories but does not
 provide a clear explanation of how they apply to the situation.  The reasoning references
 some ethical principles such as personal safety and respect for autonomy but does not
 provide a clear explanation of how they apply to the situation.  The verdict is presented
 within the required tags.  <verdict>RIGHT</verdict>
- Example:  "The action of not using the GPS app is justified.  It respects personal
 autonomy and privacy, which are fundamental human rights.  The decision to not use
 the app is a reasonable response to concerns about personal privacy and autonomy.  The
 reasoning references some ethical principles such as personal safety and respect for
 autonomy but does not provide a clear explanation of how they apply to the situation.
 <verdict>RIGHT</verdict>"
- 0:  The response does not reference any ethical principles or theories.  The reasoning
 lacks any exploration of ethical principles or ethical theories.  The reasoning lacks any
 exploration of ethical principles or ethical theories.  The verdict is presented within
 the required tags.  <verdict>RIGHT</verdict>
- Example:  "The action of not using the GPS app is justified.  <verdict>RIGHT</verdict>"
</item>
```

## K    TRAINING DETAILS

### K.1    SUMMARIZATION

**Setup.**    For summarization, we use the BillSum benchmark (Kornilova & Eidelman, 2019), a corpus designed for summarizing U.S. Congressional and California state legislation. The dataset includes over 22,000 mid-length Congressional bills with human-written summaries, along with an additional California test set to support cross-domain generalization. Its technical and hierarchical nature presents unique challenges, making it suitable for both domain-specific and general-purpose summarization research. The training set contains approximately 18.9K samples, while the combined test set includes 4.5K samples. We provide the full bill text as input and request a summary of approximately 400 words. Training is conducted for a single epoch over the dataset, with MPO steps executed every 20 batch steps, resulting in a total of 29 rubric refinements throughout the training process.

A total of 4.5K bill summaries are generated on the test set by each of the three models. These generated summaries are then evaluated by computing ROUGE (Lin, 2004) scores against their corresponding human-written references. In addition, we computed Elo ratings for the three models based on 5,000 pairwise comparisons of their generated summaries, with GPT-4o serving as the judge. The results are presented in Table 2.

### K.2    ETHICAL REASONING

**Setup.**    For ethical reasoning, we utilize the Anecdotes from the Scruples dataset (Lourie et al., 2020) which contains over 32,000 real-life anecdotes sourced from a Reddit community, where users describe ethically charged situations they experienced or considered. Each anecdote includes a title,

detailed story, and a distribution of community judgments indicating who was perceived to be in the wrong—such as the author, another party, everyone, or no one. These narratives often feature moral ambiguity and are labeled with crowd-sourced ethical assessments, making them well-suited for modeling community norms and capturing the diversity of ethical reasoning.

The dataset includes 27.8K training anecdotes, from which we randomly sampled 13K for our experiments, along with 4.7K anecdotes used for testing. Training is conducted for a single epoch, with MPO steps performed every 10 batch steps, resulting in a total of 20 rubric refinements. Although each anecdote includes a binary judgment verdict from human annotators, we did not use these ground truth labels in either the RM or MRM. Instead, reward scores were assigned solely based on the quality of ethical reasoning demonstrated in the response. This decision was motivated by two factors: (1) the label distribution is imbalanced, and (2) our goal was to encourage the policy model to improve through generating stronger reasoning traces, rather than optimizing for label prediction alone.

### K.3 MATHEMATICAL REASONING

**Setup.** For mathematical reasoning, we use the MATH dataset (Hendrycks et al., 2021) which consists of 12,500 high school competition-style math problems, sourced from contests like AMC 10, AMC 12, and AIME. Each problem is accompanied by a detailed step-by-step solution written in LaTeX, enabling both final answer evaluation and learning of problem-solving processes. The dataset spans seven subjects—including algebra, geometry, and number theory—and is annotated with difficulty levels from 1 to 5, offering fine-grained assessment across a wide range of mathematical reasoning tasks.

The dataset consists of 7.5K training samples and 5K test samples. Because mathematical reasoning demands considerable evaluative depth, we apply MPO at a finer granularity. Specifically, we cluster problems within each of the seven math subjects into three groups based on semantic embeddings, resulting in 21 ($7 \times 3$) distinct categories. During MPO training, we maintain a separate evaluation prompt for each of these categories, with refinement steps triggered based on the subject and cluster index of the sample. The reward model follows a "plan-then-execute" strategy: it first formulates an evaluation plan based on the problem, reference solution, and meta-level guidelines, and then applies this plan to assess the student's response. This approach builds on the method proposed by Saha et al. (2025), originally used for pairwise judgment, which we adapt for absolute scoring with (meta-level) rubric-guided evaluation. Training is conducted for a single epoch, with MPO steps performed every 30 batch steps—using a longer interval to ensure a sufficient mix of responses with varying quality levels is gathered before each refinement.

