# OpenReview forum: "Toward Evaluative Thinking: Meta Policy Optimization with Evolving Reward Models"
_ICLR.cc/2026/Conference — Submitted to ICLR 2026_

### Official Review · Reviewer_GHuQ · 2025-10-16

**Soundness:** 2
**Presentation:** 2
**Contribution:** 2
**Rating:** 2
**Confidence:** 4

**Summary:**

The paper proposes Meta Policy Optimization (MPO), which augments standard RLAIF by adding a meta‑reward model that periodically refines the reward model’s evaluation prompt/rubric using recent training context. This evolving rubric aims to reduce reward hacking, lessen manual prompt engineering, and provide more discriminative rewards over time. The method is instantiated with small policy models and larger RM/MRM pairs, and evaluated across essay writing, summarization, ethical reasoning, and math reasoning. Results show consistent gains over fixed‑prompt PPO baselines and robustness against typical reward‑hacking failures; on essay writing, MPO surpasses even heavily hand‑engineered “oracle” prompts under similar compute.

**Strengths:**

Practicality: Turning reward design into an evolving rubric reduces brittle prompt engineering and mitigates fixed‑prompt reward hacking.

Generality: The framework is task‑agnostic and plugs into PPO‑style RLHF/RLAIF pipelines; evaluations span writing, summarization, ethics, and math.

Evidence of robustness: Qualitative/quantitative analyses show MPO detecting and correcting gaming behaviors (e.g., title‑only outputs), while fixed‑prompt PPO can collapse.

Analysis: Tracks rubric growth and stricter scoring over iterations, explaining how finer evaluation criteria can yield more informative gradients.

**Weaknesses:**

1. Model scaling: It is unclear why experiments focus on Qwen2‑1.5B and Llama‑3.1‑8B only; a systematic sweep across Qwen2.5 scales (e.g., 0.5B→7B) under matched setups would better reveal scaling trends and improvement curves.

2. Benchmarks: Important, contemporary alignment/reasoning suites are missing; adding AlpacaEval 2.0 (length‑controlled) and Arena‑Hard variants would strengthen generalization and robustness claims.

3. Role mapping: The conceptual mapping between reward models and LLM‑as‑judge is muddled. Recent work often finds LLM‑as‑judge competitive or superior to trained reward models; the “student/junior/senior instructor” analogy would be clearer if the policy is the student, the LLM‑as‑judge the instructor/judge, and the reward model a distilled proxy of that judgment. Clarify roles and justify terminology.

4. Objective choice: The training objective is under‑motivated. Given a focus on reward shaping and rubric evolution, comparisons with DPO or GRPO would isolate whether gains come from MPO itself or from PPO specifics.

5. Baselines: Please add strong recent verifiable‑reward pipelines (e.g., RLVR‑style systems) and widely used public suites for instruction‑following/reasoning. Self‑configured evaluations are valuable but less convincing without head‑to‑head comparisons against recognized baselines.

**Questions:**

See weakness section.

---

> ### Author Response · Authors · 2025-12-03
>
> Thank you for your time in reviewing our work. Below, we provide our responses.
>
> ---
>
> > Model scaling: It is unclear why experiments focus on Qwen2‑1.5B and Llama‑3.1‑8B only; a systematic sweep across Qwen2.5 scales (e.g., 0.5B→7B) under matched setups would better reveal scaling trends and improvement curves.
>
> This limitation was primarily due to the resource constraints we faced at the time, as a full sweep would require exploring many combinations of policy, student, RM, and MRM configurations. We are currently conducting these parameter-sweep experiments and plan to include the results in the next revision. While such scaling results would certainly be informative, we do not believe their absence undermines the core findings or the demonstrated effectiveness of our approach, and we hope the reviewer will not view this as a major weakness.
>
> > Benchmarks: Important, contemporary alignment/reasoning suites are missing; adding AlpacaEval 2.0 (length‑controlled) and Arena‑Hard variants would strengthen generalization and robustness claims.
>
> Our initial goal was to evaluate the feasibility of meta-evaluation within RL training, which led us to train models on task-specific domains rather than developing a general-purpose LLM. That said, we agree that standardized evaluations remain valuable. Accordingly, we conducted additional experiments using the Arena-Hard 2.0 benchmark. Since this benchmark includes samples from the "coding," "math," and "creative writing" domains, and our focus during the rebuttal period was on creative-writing capabilities, we translated all creative-writing prompts into English and evaluated our models on that subset.
>
>
> Judged with GPT-5:
>
> | Model         | Scores (%) | CI (%) |
> |---------------|----------------|-----------|
> | PPO     |    7.5  | (-1.2 / +1.2) |
> | MPO-PPO |    8.2  | (-1.1 / +1.4) |
> | GRPO    |    6.8  | (-1.5 / +1.5) |
> | MPO-GRPO|    8.7  | (-1.2 / +1.2) |
>
>
> > Role mapping: The conceptual mapping between reward models and LLM‑as‑judge is muddled. Recent work often finds LLM‑as‑judge competitive or superior to trained reward models; the “student/junior/senior instructor” analogy would be clearer if the policy is the student, the LLM‑as‑judge the instructor/judge, and the reward model a distilled proxy of that judgment. Clarify roles and justify terminology.
>
> In our framework, there are two types of LLM-based reward models: the Reward Model (RM) and the Meta Reward Model (MRM). As you noted, the policy serves as the student and the RM functions as the LLM-as-judge. The MRM, in turn, operates as an LLM-as-meta-judge, since it evaluates and refines the judgments produced by the RM. We will clarify this role hierarchy and terminology more explicitly in the revised manuscript.
>
> > Objective choice: The training objective is under‑motivated. Given a focus on reward shaping and rubric evolution, comparisons with DPO or GRPO would isolate whether gains come from MPO itself or from PPO specifics.
>
> This is a valid point. During the rebuttal period, we implemented MPO within the GRPO framework to better disentangle the effects of MPO from PPO-specific factors. Our integration involved the following steps:
> - GRPO-specific modifications
>   - We preserved the original GRPO training code and simply replaced the reward computation with the RM-derived reward.
>   - We inserted MPO steps on top of the existing GRPO training loop.
> - Sample selection
>   - Because GRPO generates multiple rollouts per sample, we needed a more informed strategy for selecting which samples to use for MPO updates.
>   - To achieve this without incurring additional LLM calls, we embedded all candidate samples using a recent embedding model and applied PCA to identify the most representative samples for MPO.
>
>
> | Model         | Mean Elo Score | Std. Dev. |
> |---------------|----------------|-----------|
> | MPO-GRPO (Qwen2-1.5B, 32B_32B)   | 1020.233       | 6.472     |
> | Plain GRPO (Qwen2-1.5B, 32B) | 966.811        | 11.803    |
>
>
> > Baselines: Please add strong recent verifiable‑reward pipelines (e.g., RLVR‑style systems) and widely used public suites for instruction‑following/reasoning. Self‑configured evaluations are valuable but less convincing without head‑to‑head comparisons against recognized baselines.
>
> Thank you for the suggestion. As noted above, we incorporated GRPO and its MPO-enhanced variant to address your point about including a more recent RLVR-style baseline. We also evaluated our models on selected domains of the Arena-Hard v2.0 benchmark and found that MPO consistently provides additional improvements in those settings.

---

### Official Review · Reviewer_S4JR · 2025-11-01

**Soundness:** 3
**Presentation:** 3
**Contribution:** 3
**Rating:** 6
**Confidence:** 4

**Summary:**

The paper proposes Meta Policy Optimization (MPO) for the RLAIF setting, inspired by the psychological concept of metacognition. In this framework, a meta-reward model (MRM) dynamically refines the reward model’s prompt throughout training. The MRM monitors the evolving training landscape by processing the prompt instructions, reference solutions (if available), policy generations, the reward model’s evaluation rubric, and the scores assigned to those generations. Using this information, the MRM identifies weaknesses in the current rubric that the policy may be exploiting (or is likely to exploit) and modifies the rubric to make it more targeted and fine-grained. This helps the reward model resist reward hacking by the policy and promotes more stable policy optimization. MPO reduces the need for manual prompt design and proves effective across diverse tasks without requiring specialized reward designs. Experimentally, the authors show that MPO outperforms PPO with expert or auto prompting across four different domains.

**Strengths:**

The paper proposes Meta Policy Optimization (MPO) for the RLAIF setting, inspired by the psychological concept of metacognition. MPO addresses reward hacking in RLAIF by introducing a meta-reward model (MRM) that periodically refines the reward model’s prompt during training. This ensures that the evaluation rubric remains granular, targeted, and resistant to exploitation by the policy. MPO is a timely contribution toward mitigating reward hacking in RLAIF and promotes more stable policy optimization. Furthermore, the prompts used for the MRM are general and task-agnostic, enabling their usage across diverse domains.

MPO demonstrates strong effectiveness compared to approaches that rely on static, hand-crafted prompts, even those designed by domain experts, across diverse tasks such as essay writing, summarization, ethical reasoning, and mathematical reasoning, showcasing its versatility. Additionally, MPO reduces the burden of prompt engineering by automatically refining the reward model’s prompt throughout training based on the observed training context.

Finally, the paper is clearly written and well-organized, making it easy to follow.

**Weaknesses:**

The sample selection process for rubric refinement is completely random. Samples drawn from much earlier stages of training may not be informative, as the training context and policy behavior could have evolved significantly. Given this, it may be more effective to prioritize recent samples or those with higher estimated informativeness when updating the rubric. Such an approach could make the refinement process more adaptive to the model’s current failure modes.

As the MRM continuously evolves the reward model’s rubric, the rubric appears to become increasingly complex and fine-grained over the course of training. This process resembles inferring highly detailed reward functions that fit the observed training context but may not generalize well to unseen or downstream tasks. In light of this, it might be useful to regularize the inferred rubric, for instance, by penalizing excessive complexity or enforcing smoothness constraints, to improve generalization and stability.

Another concern is that the scoring scale of the rubric can change dynamically during training. At one point, the maximum score might be 30, whereas at a later stage it could increase to 50. This variability may lead to inconsistent reward magnitudes for the policy. To address this, it would make sense to use a normalized reward score, $s\in [0,1]$, ensuring a consistent and comparable reward scale across training iterations.

Finally, the experiments are primarily conducted on smaller models (e.g., Qwen2-1.5B-Instruct), with limited evaluation on larger LLMs due to resource constraints. This leaves open questions about scalability, in particular, whether MPO remains effective and stable as model size increases and training dynamics become more complex.

**Questions:**

1) Does the rubric becoming more complex over the course of training, as it is evolved by the MRM, affect the generalization performance of the LLM aligned via MPO? Wouldn't regularizing the rubric help improve generalization without sacrificing MPO's ability to curtail reward hacking?

2) Since the rubric scoring scale can change over the course of training, wouldn't it be better to use a normalized score as the training signal for the RL algorithm?

3) Do you have experimental results for other model scales (3B, 7B, 13B, etc.) and potentially other models (e.g., LLaMA) for the policy?

4) In Section 3.3.1, PPO-aligned 32B_AP receives the highest rating in pairwise Elo evaluations. The hypothesis was that the GPT-4o judge favors outputs from models aligned using evaluation rubrics it helped generate. Why was this not observed in the results for the essay writing task?

5) Why were 72B RM and MRM sizes used only for the essay writing task and not for the other three domains?

---

> ### Author Response · Authors · 2025-12-03
>
> Thank you for your time and effort in reviewing our paper and acknowledging the effectiveness of our approach. We provide our responses below.
>
> ---
>
> > The sample selection process for rubric refinement is completely random. ... it may be more effective to prioritize recent samples or those with higher estimated informativeness when updating the rubric. Such an approach could make the refinement process more adaptive to the model’s current failure modes.
>
> Thank you for raising this point. We agree with your assessment. During the rebuttal period, we developed a more principled sample-selection strategy for the MPO steps, which turned out to be crucial for making MPO work effectively within the GRPO framework. Because GRPO produces multiple rollouts per sample, we needed a more informed method for identifying meaningful rollouts.
>
> Our initial attempt involved fast screening via LLM prompting, but the screening quality was insufficient, and more careful prompting would have resulted in excessive LLM calls and prohibitive overhead. Ultimately, we adopted a more scalable approach: we used a recent embedding model (https://github.com/QwenLM/Qwen3-Embedding) to embed the candidate samples and applied PCA to select the top N (=20) samples most representative of the aggregated batch. This method yielded more stable and effective policy training under the GRPO setting.
>
> We acknowledge that there are likely more sophisticated or adaptive strategies for sample selection that we have not yet explored, and we view this as a promising direction for future work.
>
> > As the MRM continuously evolves the reward model’s rubric, the rubric appears to become increasingly complex and fine-grained over the course of training. This process resembles inferring highly detailed reward functions that fit the observed training context but may not generalize well to unseen or downstream tasks. In light of this, it might be useful to regularize the inferred rubric, for instance, by penalizing excessive complexity or enforcing smoothness constraints, to improve generalization and stability.
>
> The current scope of the paper is to examine each downstream task individually, with varying levels of "evaluative thinking," in order to first assess the feasibility of our approach. Within a given downstream task, we observed that when the evaluation required attention to an unseen aspect (e.g., identifying missing "evidence" in argumentative essays), the MRM was able to detect this gap and introduce a new criterion accordingly.
>
> However, we agree with your broader point: if the goal were to build a general-purpose LLM capable of handling diverse tasks, an additional layer of rubric management would be necessary. Such a system would need to select, adapt, or construct the appropriate rubric for each downstream task, potentially with mechanisms to regularize or control rubric complexity to support generalization and stability.
>
> > scoring scale of the rubric can change dynamically during training. use a normalized reward score ...
>
> We thank you for raising this point. Incorporating your feedback, we added an option to normalize the total reward to the [0, 1] range and confirmed that this leads to more stable training.
>
> > The experiments are primarily conducted on smaller models (e.g., Qwen2-1.5B-Instruct), with limited evaluation on larger LLMs due to resource constraints.
>
> In Appendix E, we present additional results using a larger policy model from a different LLM family, microsoft/Phi-3-mini-128k-instruct (4B), together with a much smaller (M)RM (Llama-3.1-8B-Instruct) for the MPO steps. Even under this configuration, we observed consistent improvements over standard PPO training. We believe that, as noted in the previous question, as long as the MRM can reliably identify inconsistencies in rubric evaluations or areas for improvement, MPO should yield performance gains. The MRM’s effectiveness in doing so naturally depends on its underlying model family and capabilities. We are currently conducting parameter-size sweep experiments and plan to include the results in our next revision.
>
> > ... Wouldn't regularizing the rubric help improve generalization without sacrificing MPO's ability to curtail reward hacking?
>
> We view the introduction of stricter criteria and more detailed scoring schemes as a meaningful form of regularization in itself. Rather than overfitting to a particular task, these refinements provide more comprehensive and structured guidance for evaluating that task. As noted above, extending this approach to multiple task domains is beyond the scope of the current paper and would indeed require an additional layer of rubric management, as you suggested.
>
> > ... wouldn't it be better to use a normalized score as the training signal for the RL algorithm?
>
> Adopting your comment, we normalized the total reward to fall within 0 to 1 range.

---

> > ### Author Response · Authors · 2025-12-03
> >
> > > Do you have experimental results for other model scales (3B, 7B, 13B, etc.) and potentially other models (e.g., LLaMA) for the policy?
> >
> > In Appendix E, we present additional results using a larger policy model from a different LLM family, microsoft/Phi-3-mini-128k-instruct (4B), together with a much smaller (M)RM (Llama-3.1-8B-Instruct) for the MPO steps. Even under this configuration, we observed consistent improvements over standard PPO training. We believe that, as noted in the previous question, as long as the MRM can reliably identify inconsistencies in rubric evaluations or areas for improvement, MPO should yield performance gains. The MRM’s effectiveness in doing so naturally depends on its underlying model family and capabilities. We are currently conducting parameter-size sweep experiments and plan to include the results in our next revision.
> >
> > > In Section 3.3.1, PPO-aligned 32B_AP receives the highest rating in pairwise Elo evaluations. The hypothesis was that the GPT-4o judge favors outputs from models aligned using evaluation rubrics it helped generate. Why was this not observed in the results for the essay writing task?
> >
> > We believe you are referring to the known tendency of LLM judges to prefer responses that resemble their own style. While the rubric for the _AP model was indeed generated by GPT-4o, that rubric is relatively too simple (as can be seen in the anonymized code repository) to project any GPT4-like writing style. More importantly, in this evaluation setup, the GPT-4o judge is assessing outputs from non-GPT policy models rather than comparing its own generations to those of another model. As a result, the usual self-preference bias is less likely to manifest in this setting.
> >
> > > Why were 72B RM and MRM sizes used only for the essay writing task and not for the other three domains?
> >
> > This was purely for efficiency reasons. After examining the results on the essay-writing task, we found that 32B models were sufficient (and substantially faster) while still outperforming the baselines. Therefore, we used the 32B configurations for the remaining tasks.

---

### Official Review · Reviewer_d9Cr · 2025-11-01

**Soundness:** 2
**Presentation:** 2
**Contribution:** 2
**Rating:** 2
**Confidence:** 4

**Summary:**

The paper proposes Meta Policy Optimization (MPO), a framework that dynamically refines the evaluation rubric used by a reward model (RM) during reinforcement learning from AI feedback (RLAIF). The core idea is to introduce a meta-reward model (MRM) that periodically analyzes the RM’s scoring behavior and updates its rubric to mitigate reward hacking and reduce reliance on manual prompt engineering. The authors evaluate MPO on four tasks—essay writing, summarization, ethical reasoning, and mathematical reasoning—reporting improvements over fixed-prompt baselines, including hand-crafted and AutoPrompt-generated rubrics.

**Strengths:**

- The motivation—addressing reward hacking and prompt brittleness in RLAIF—is well-articulated and practically relevant.
- The MPO framework is conceptually clean and integrates naturally into existing PPO-based pipelines.

**Weaknesses:**

1. **Limited experimental scope and reliability**: The evaluation is conducted exclusively on a single small policy model (Qwen2-1.5B) and only with PPO. This raises concerns about the generalizability of the findings. In the current RLAIF literature, standard benchmarks such as **Arena-Hard-v2** or **Alpaca-Eval** are expected for alignment claims, yet these are entirely absent. Without results on more representative models (e.g., 7B+ scale) or alternative RL algorithms (e.g., GRPO), it is unclear whether MPO’s benefits are robust or merely artifacts of a narrow setup.

2. **Rubric design appears misaligned with task heterogeneity**: The paper implies that a single evolving rubric is shared across all queries within a task (e.g., all essay prompts use one rubric).  However, it is natural that different queries may require distinct evaluation criteria (e.g., creativity vs. factuality in essays). The current design risks oversimplifying the complexity of human preferences.

3. **Oracle rubric and evaluation protocol lack rigor**: The “oracle” rubric is derived from 60+ PPO runs on the same small model—an ad hoc and non-standard baseline. More convincingly, the paper could compare MPO’s evolved RM against a much stronger fixed judge (e.g., Qwen-2507-235B or GPT-4o) to assess whether dynamic rubric refinement truly closes the gap with top-tier static evaluators. As it stands, the claim that MPO “surpasses human-engineered prompts” is overstated given the weak oracle baseline and reliance on GPT-4o as the sole judge (which may favor its own prompt styles).

**Questions:**

seed weakness

---

> ### Author Response · Authors · 2025-12-03
>
> Thank you for your time in reviewing our work. Below, we provide our responses.
>
> ---
>
> > Limited experimental scope and reliability: The evaluation is conducted exclusively on a single small policy model (Qwen2-1.5B) and only with PPO. This raises concerns about the generalizability of the findings. In the current RLAIF literature, standard benchmarks such as Arena-Hard-v2 or Alpaca-Eval are expected for alignment claims, yet these are entirely absent. Without results on more representative models (e.g., 7B+ scale) or alternative RL algorithms (e.g., GRPO), it is unclear whether MPO’s benefits are robust or merely artifacts of a narrow setup.
>
> In Appendix E, we present additional results using a larger policy model from a different LLM family, microsoft/Phi-3-mini-128k-instruct (4B), together with a much smaller (M)RM (Llama-3.1-8B-Instruct) for the MPO steps. Even under this configuration, we observed consistent improvements over standard PPO training. We believe that, as noted in the previous question, as long as the MRM can reliably identify inconsistencies in rubric evaluations or areas for improvement, MPO should yield performance gains. The MRM’s effectiveness in doing so naturally depends on its underlying model family and capabilities. We are currently conducting parameter-size sweep experiments and plan to include the results in our next revision.
>
> Reflecting your valuable feedback, we also present additional results on Arena-Hard-v2 benchmark where we appropriately selected samples for creative writing domain. We translated all creative-writing prompts into English and evaluated our models on that subset.
>
>
> Judged with GPT-5:
>
> | Model         | Scores (%) | CI (%) |
> |---------------|----------------|-----------|
> | PPO     |    7.5  | (-1.2 / +1.2) |
> | MPO-PPO |    8.2  | (-1.1 / +1.4) |
> | GRPO    |    6.8  | (-1.5 / +1.5) |
> | MPO-GRPO|    8.7  | (-1.2 / +1.2) |
>
>
>
> > Rubric design appears misaligned with task heterogeneity: The paper implies that a single evolving rubric is shared across all queries within a task (e.g., all essay prompts use one rubric). However, it is natural that different queries may require distinct evaluation criteria (e.g., creativity vs. factuality in essays). The current design risks oversimplifying the complexity of human preferences.
>
> We agree with the reviewer's intuition that a single rubric per task may not be insufficient, as even within a nominal task category (e.g., math), the underlying query types can vary substantially. As you note, algebra-focused math problems differ markedly from proof-style questions. A monolithic rubric risks collapsing these heterogeneous preference dimensions into an overly coarse evaluation signal.
>
> In our experiments, we already address this issue in the math domain by sub-clustering the dataset and maintaining separate evaluation prompts (rubrics) for each cluster. Specifically, we use subject-based clusters and initialize a distinct MRM-generated rubric for each, with MPO-based refinements applied every 30 batch steps (see line 395 of the draft). Empirically, this yields better stability and generalization, particularly in domains where evaluative reasoning is deeper and more instance-specific.
>
> By contrast, in tasks whose evaluation relies more on overarching, breadth-oriented criteria, such as essay writing, we found that a single evolving rubric was sufficient. When a new criterion (e.g., "creativity") becomes necessary for a particular query and is absent from the current rubric, the MRM reliably identifies this gap and explicitly adds the missing criterion during refinement.
>
> Looking ahead, we envision incorporating an additional layer that manages multiple rubrics across tasks and domains. We hope this clarification addresses the reviewer's concern.

---

> > ### Author Response · Authors · 2025-12-03
> >
> > > Oracle rubric and evaluation protocol lack rigor: The "oracle" rubric is derived from 60+ PPO runs on the same small model—an ad hoc and non-standard baseline. More convincingly, the paper could compare MPO’s evolved RM against a much stronger fixed judge (e.g., Qwen-2507-235B or GPT-4o) to assess whether dynamic rubric refinement truly closes the gap with top-tier static evaluators. As it stands, the claim that MPO “surpasses human-engineered prompts” is overstated given the weak oracle baseline and reliance on GPT-4o as the sole judge (which may favor its own prompt styles).
> >
> > We used the term "oracle" to refer to the final outcome of extensive human prompt-engineering tailored to this specific policy model and dataset; essentially the best-performing prompt that avoided reward hacking after numerous PPO training runs conducted "in advance." In this sense, it serves as a meaningful reference point, demonstrating that MPO can bypass the many failed PPO attempts typically required to arrive at such a prompt.
> >
> > We agree that comparing against a much stronger, large-scale RM would offer additional insight, but this constitutes only one component of our overall evaluation setup. We do not believe the absence of such a comparison renders the evaluation protocol lacking in rigor. Rather, the oracle baseline is intended to capture the practical endpoint of human-engineered reward shaping for the given setting, and our results show that MPO can effectively surpass this without manual intervention.

---

### Official Review · Reviewer_VSwY · 2025-11-08

**Soundness:** 3
**Presentation:** 2
**Contribution:** 3
**Rating:** 6
**Confidence:** 4

**Summary:**

This paper introduces Meta Policy Optimization (MPO), a novel framework that tackles two persistent challenges in Reinforcement Learning from AI Feedback (RLAIF) for large language models: vulnerability to reward hacking, where models exploit flaws in the reward signal, and the heavy reliance on brittle, manual prompt engineering for reward models. Inspired by metacognition and evaluative thinking, MPO augments the standard RLAIF pipeline with a meta-reward model (MRM) that dynamically refines the reward model's prompt throughout training. Empirically, MPO demonstrates significant advantages across diverse tasks spanning the depth-breadth spectrum of evaluative thinking, such as essay writing, summarization, ethical reasoning, and mathematics. It achieves performance on par with or superior to models using extensively hand-crafted prompts, while crucially preventing policy collapse due to reward hacking, as observed in fixed-prompt setups.

**Strengths:**

- MPO directly addresses two of the most significant pain points in RLAIF: reward hacking and the immense burden of manual prompt engineering.
-  A major strength is the extensive empirical validation across four distinct tasks, each representing different challenges on the spectrum of evaluative thinking.
- The paper goes beyond simply reporting results to analyze how MPO works. The discussion on the evolution of the rubric's linguistic structure provides valuable insights into the framework's inner workings.

**Weaknesses:**

- The entire MPO process hinges on the quality of the MRM's analysis and refinements. The paper does not deeply explore what happens if the MRM itself is flawed, generates poor rubrics, or introduces new biases.
- While the paper reports an 11% compute overhead and argues it is modest, this is a critical factor for adoption.

**Questions:**

- The paper demonstrates strong results on single-turn generation tasks. How would you envision and potentially adapt the MPO framework for multi-turn interactive tasks, such as dialogue or long-horizon instruction following?
- Your results show that performance improves with the size of both the RM and MRM. Could you discuss the interplay between the policy model size and the effectiveness of MPO? What are the optimal scaling relationships between these three components?

---

> ### Author Response · Authors · 2025-12-03
>
> Thank you for your time and effort in reviewing our manuscript, and for recognizing the strengths of our approach. Below, we provide our responses to your feedback.
>
> ---
>
> > The entire MPO process hinges on the quality of the MRM's analysis and refinements. The paper does not deeply explore what happens if the MRM itself is flawed, generates poor rubrics, or introduces new biases.
>
> This is an important point. Our approach assumes that the MRM is of sufficiently high quality, and that the processes of identifying issues within solutions and proposing refinements are inherently less error- or bias-prone than generating them. Although a full investigation of failure modes is outside the scope of this work, one potential mitigation is to aggregate or cross-validate analyses produced by multiple MRMs, which could help reduce individual model biases or errors. We will mention this point in our manuscript.
>
> > While the paper reports an 11% compute overhead and argues it is modest, this is a critical factor for adoption.
>
> The computational overhead can be reduced in two ways: (1) allocating more resources to reward computation, and (2) performing MPO updates less frequently by using a larger and better-curated sample pool. In our manuscript, we used a 1 × H100 setup for reward computation and 3 × H100s for RL training. In retrospect, a 2:2 allocation would likely have resulted in faster overall training. Moreover, in follow-up experiments (see our other responses) using MPO within the GRPO framework, we found that substantially fewer MPO steps are required when sample selection is performed systematically rather than uniformly at random. Based on these factors, we expect that the computational overhead can be reduced to below 5%.
>
> > The paper demonstrates strong results on single-turn generation tasks. How would you envision and potentially adapt the MPO framework for multi-turn interactive tasks, such as dialogue or long-horizon instruction following?
>
> That is an excellent direction for further research. We believe the core concept of meta-evaluation naturally extends to multi-turn tasks: when there is an overarching objective and a senior LLM can observe the full interaction trajectory (including the student model’s responses, the junior model’s responses, and intermediate human inputs) it can similarly identify failures, missed opportunities, or areas for refinement. Adapting MPO to this setting would require supplying the additional conversational context (e.g., intermediate human turns) during RL training, but the overarching oversight and improvement mechanism would remain essentially the same.
>
> > Your results show that performance improves with the size of both the RM and MRM. Could you discuss the interplay between the policy model size and the effectiveness of MPO? What are the optimal scaling relationships between these three components?
>
> In Appendix E, we present additional results using a larger policy model from a different LLM family, microsoft/Phi-3-mini-128k-instruct (4B), together with a much smaller (M)RM (Llama-3.1-8B-Instruct) for the MPO steps. Even under this configuration, we observed consistent improvements over standard PPO training. We believe that, as noted in the previous question, as long as the MRM can reliably identify inconsistencies in rubric evaluations or areas for improvement, MPO should yield performance gains. The MRM’s effectiveness in doing so naturally depends on its underlying model family and capabilities. We are currently conducting parameter-size sweep experiments and plan to include the results in our next revision.

---

### Author Response · Authors · 2025-12-03

We thank all reviewers for their time and for providing valuable feedback on our manuscript.

---

We are encouraged to see that all reviewers agree on both the motivation and the empirical effectiveness of our approach. The main concerns raised across reviews can be summarized as follows:
1. The need for an additional test suite for general preference alignment.
2. The need for an additional RL framework, preferably GRPO.
3. The need for parameter-sweeping experiments for RMs and MRMs.

---

During the rebuttal period, we addressed the first two points.
First, we provided evaluation results for both our approach and the baseline methods on the Arena-Hard v2.0 benchmark for creative writing tasks, demonstrating that the improvements observed with MPO are consistent across settings.

Second, we implemented MPO within the GRPO framework. Specifically, we retained most of the original GRPO pipeline but replaced the reward function with calls to the RM, and incorporated MPO steps into the training loop. Again, performing meta-evaluation during GRPO training improves performance and yields a more fine-grained rubric.

We are yet to provide results for the third point, as it requires hefy computational resources but we would like to point out that we do provide experimental results on different combination of LLM model family and pramemter sizes in Appendix E.

We would also like to highlight that, to the best of our knowledge, we were the first to conceptualize the notion of meta-evaluation via an evolving rubric. Since then, multiple research teams have independently explored similar ideas and reported improvements, further validating the usefulness of this direction:
- DR Tulu: Reinforcement Learning with Evolving Rubrics for Deep Research
https://arxiv.org/abs/2511.19399
- Rubrics as Rewards: Reinforcement Learning Beyond Verifiable Domains
https://arxiv.org/abs/2507.17746

We hope that our additional experiments and clarifications adequately address the reviewers’ concerns, and we believe they further strengthen the contributions of our work.

---

### Meta-Review · Area_Chair_m8q8 · 2026-01-03

**Summary:**

This is a boarderline paper and the reviewers' scores are quite diverse (2,2,6,6). Most reviewers appreciate the importance and the potential impact of this topic about preventing reward hacking. The main concerns (Reviewer d9Cr, Reviewer GHuQ) are about the whether the proposed method can scale beyond the small models tested in the paper (Qwen2-1.5B and only with PPO). In addition the lack of benchmarks like Arena-Hard v2.0 was raised. In the rebuttal, the authors include extra experiments, using microsoft/Phi-3-mini-128k-instruct (4B) and GRPO, and creative writing from Arena-Hard v2.0. The reviewers also raise concern on the meta reward may be flawed, and may not be able to capture and revise the RM's prompts well by using a single rubric for all queries.

I think all these are legit concerns, though I think the authors have tried to address them. While most concerns in my mind are addressed within the scale of comparison adopted in the paper, the remaining issue mainly is whether the experimental results cover large enough models, like 7B ones, which can demonstrate different behaviors. This is one of the main concerns shared by the negative reviewers.

While I appreciate the novelty here, I agree that larger LMs are needed, as they have more capacity to actually reward hack. It is unclear whether the ideas here would work based on only the results of the smaller LM policies. The results here show improved performance, but it is unclear whether the motivation of "vulnerability to reward hacking" is indeed addressed systematically.

**Reviewer Concerns:**

See summary.

**Reviewer Scores:**

The negative reviewers might increase their slightly, but I think they will remain negative most likely. This leads a boarderline situation.

---

### Decision · Program_Chairs · 2026-01-26

Reject